# Modeling heterogeneous signaling dynamics of macrophages reveals principles of information transmission in stimulus responses

Xiaolu Guo [1,2], Adewunmi Adelaja [1,2,4], Apeksha Singh [1,2], Roy Wollman [1,3] & Alexander Hoffmann [1,2] ✉

Macrophages initiate pathogen-appropriate immune responses with the activation dynamics of transcription factor NFκB mediating specificity. Live-cell imaging revealed the stimulus-response specificity of NFκB dynamics among populations of heterogeneous cells. To study stimulus-response specificity beyond what is experimentally accessible, we develop mathematical model simulations that capture the heterogeneity of stimulus-responsive NFκB dynamics and the stimulus-response specificity performance of the population. Complementing experimental data, extended-dose response simulations improved channel capacity estimates. By collapsing parameter distributions, we locate information loss to receptor modules, while the negative-feedback-containing core module shows remarkable signaling fidelity. Further, constructing virtual single-cell networks reveals the stimulus-response specificity of single cells. We find that despite stimulus-response specificity limitations at the population level, the majority of single cells are capable of responding specifically to immune threats, and that the few instances of stimulus-pair confusion are highly uncorrelated. The diversity of blindspots enable small consortia of macrophages to achieve perfect stimulus distinction.

As a first-line defense against invading pathogens, immune sentinel cells can sense pathogen-associated molecular patterns (PAMPs), damage-associated molecular patterns (DAMPs), or cytokines produced by first responders. The information from stimuli is captured by pattern recognition receptors (PRRs) or cytokine receptors and transmitted to a signaling network whose effectors determine cellular functions[1–4]. Upon stimulation, immune sentinel cells initiate cell-intrinsic defenses, local immune activity, and systemic immune activation. Immune activation responses must be tailored to the specific immune challenges, in ensure an effective immune response while

avoiding unnecessary activation of harmful immune effectors. This suggests that stimulus-response-specificity (SRS), the ability of cells to mount stimulus-specific immune responses, is a functional hallmark of healthy immune sentinel cells[4–6].

The SRS of immune sentinel cell effectors has been characterized at the level of gene expression responses[7], which are thought to be controlled by combinatorial and temporal coding of signaling pathways[4,5,8]. One of such signaling pathway, the transcription factor NFκB, is highly dynamic as revealed by early biochemical and imaging studies[9–13]; furthermore, these dynamics are stimulus-specific[14,15] and

[1]Institute for Quantitative and Computational Biosciences, University of California Los Angeles, Los Angeles, CA, USA. [2]Department of Microbiology, Immunology, and Molecular Genetics, University of California Los Angeles, Los Angeles, CA, USA. [3]Departments of Integrative Biology and Physiology, and Chemistry and Biochemistry, University of California Los Angeles, Los Angeles, CA, USA. [4]Present address: Harvard Combined Dermatology Residency Training Program, Boston, MA, USA. ✉e-mail: ahoffmann@ucla.edu

regulate the stimulus-specificity of immune response gene expression[9,15–19].

At the single cell level however, NFκB dynamics display pronounced cell-to-cell heterogeneity[10,12,20] casting doubt on the notion that such signaling dynamics could transmit stimulus information with fidelity. Only with single-cell trajectory data available, specific dynamic features within the NFκB activation trajectories could be identified conveying information about the stimulus[16,21,22]. These informative NFκB dynamic features, termed "signaling codons" are Speed, Peak, Duration, Total Activity, Early-or-Late, and Oscillation. Machine learning classifiers validated the importance and sufficiency of signaling codons in driving SRS[21,23,24].

The SRS of macrophage signaling dynamics has thus far been characterized from experimental data that is not only unavoidably limited in the number of stimuli or doses tested, but also in the kind of perturbations that can be implemented. Further, while SRS limitations have been documented[7,21,25], it remains unclear whether assessments of mutual information for the available data may be underestimates of the true channel capacity of the pathway[26,27]. Further, for common signaling motifs, intrinsic noise was not found to limit the channel capacity[28]. Most critically, all SRS estimates thus far are at the population level as a naïve cell can only be stimulated once, necessitating different (but analogous) cell populations to assess responses to different stimulus conditions. How population level SRS relates to the specificities of individual cells (scSRS) remains unclear. While some of these biological questions driven by cellular heterogeneity may be challenging to address systematically in experiments, others, such as scSRS, may be nearly impossible. However, they could potentially be explored with a mechanistic mathematical model, if it reliably represents the biochemical regulatory mechanisms and accounts for experimental stimulus-response data (Fig. 1A).

Mathematical models (composed of Ordinary Differential Equations, ODEs) of the core NFκB signaling module have aided our understanding of the regulatory mechanisms that control NFκB dynamics via the prominent IκBα negative feedback loop[29,30]. While A20 is transcriptionally regulated by NFκB it does not regulate NFκB dynamics in the short term but provides rheostat control upstream of IKK and desensitizes the pathway to subsequent signals[31,32]. In contrast, short-term regulation of IKK is predominantly governed by non-transcriptional feedback mechanisms[33]. The IκB-NFκB signaling module receives inputs in the form of IκB kinase (IKK) activities that are first generated by ligand-specific receptor modules. Building on models of the TNF[32,34] and LPS[35] signaling modules, a mathematical model of NFκB activation in response to five pro-inflammatory ligands was recently established, adding modules for the bacterial PAMPs, Pam3CSK4 and CpG, and the viral PAMP, poly(I:C)[21]. Such mechanistic models for signaling pathways in mammalian systems, for example, not only NFκB[13,21], but p53[36,37], and MAPKs[38], codify a wealth of biochemical knowledge including the reaction network topology, cellular abundances of regulators, their half-lives, and interaction constants and provide simulations of the population average or a stimulus response that is representative of the population. Several studies explored potential mechanisms that may account for cell-to-cell heterogeneity by distributing parameters or incorporating stochasticity[11,13,35,39–42]. However, these models were not fit to and do not capture the heterogeneity observed experimentally, which limits their applicability for investigations of information transmission by heterogeneous cell populations (Fig. 1A).

Cellular heterogeneity among genetically identical cells can arise from either extrinsic or intrinsic noise[43]. Extrinsic noise is based on cell-to-cell variability in the expression of enzymes controlling key reactions but whose expression is not explicitly represented in the model, while intrinsic noise refers to the stochasticity of biochemical reactions. Extrinsic noise appears to be the primary source of biological noise during short time courses as shown by highly correlated cell

death or cell division decisions of sister cells[44,45]. For NFκB dynamics, one extrinsic noise source, endosomal maturation, was found to be an important contributor to LPS-stimulated NFκB signaling heterogeneity[35]. Another, TNF receptor abundance, impacts the oscillatory propensity of TNF-induced NFκB signaling[42,46]. However, it remains unclear whether the proposed model topology[21] coupled with extrinsic noise is sufficient to recapitulate the heterogeneous NFκB responses to multiple stimuli and account for the SRS performance that is experimentally observed in macrophage populations.

Here, we advanced the mechanistic model simulations for the NFκB signaling pathway from one cell (representative or population average) to all single cells within the population (Fig. 1A). These simulations captured not only the stimulus-specific heterogeneous NFκB dynamics observed experimentally, but also reproduced the SRS performance metrics of actual live macrophages. To achieve this, we used a nonlinear mixed-effects (NLME) model with stochastic approximation expectation maximization (SAEM) algorithm to parameterize the model from extensive, newly generated single-cell NFκB datasets. Given the reliability of this mechanistic model, it allowed for forward predictions outside the training range to investigate how cells respond to finer dose gradations and different genotypes. Furthermore, we developed a denoising strategy to locate major information loss to receptor modules of the NFκB signaling network, which is validated by previous experimental studies[33,47]. We also designed a workflow to infer the stimulus response specificities of single cells (scSRS). Our simulations reveal high diversity in scSRS profiles, enabling potentially perfect stimulus distinction of small consortia of collaborating cells. These knowledge-based mechanistic modeling studies provide insights that were not accessible by direct experimentation.

## Results
### Mathematical modeling of heterogeneous single-cell NFκB dynamics
To develop computational simulations of heterogeneous NFκB dynamics, we leveraged an established mathematical model of the signaling network[21] in response to five pro-inflammatory ligands: Tumor Necrosis Factor (TNF), and PAMPs, namely the TLR2 ligand Pam3CysSerLys4 (Pam), the TLR9 ligand Cytosine-phosphate-Guanine (CpG), the TLR4 ligand Lipopolysaccharide (LPS), the TLR3 ligand Polyinosinic:polycytidylic acid (pIC), at several doses (Fig. 1B). The published model simulation produces a single NFκB trajectory for each ligand and dose that is representative of the observed data[21]. The mechanistic model contains 52 species, 101 reactions, and 133 parameters (Supplementary Datasets 1 and 2, Eq. 1 to 52 in Supplementary Notes). It is organized into five receptor modules (Fig. 1B) with each responding to its cognate ligand. These receptor modules feed via IKK into a common core module, which comprises the NFκB-IκBα negative feedback loop.

To account quantitatively for the heterogeneous NFκB dynamics present in the population, distributions of key biochemical reactions had to be parameterized. Because PAMPs induce the production of TNF[34], the published NFκB trajectories are the result of the combined action of PAMPs and TNF signaling, especially at low doses[21]. To facilitate parameterization of each signaling module, we generated a new experimental dataset of NFκB dynamic trajectories using a TNF blocking agent when PAMP ligands are employed (Fig. S1A, Methods – Experimental data generation). The established live-cell imaging and analysis workflow enabled the generation of a quantitative dataset using five ligands (TNF, pIC, Pam, CpG, or LPS) at 3 doses (low, medium and high), with 176 to 744 single cell NFκB trajectories for each condition (Table 1).

We used the non-linear mixed-effects model (NLME) to elucidate the variations in parameters or initial species values responsible for generating heterogeneous trajectories. The ordinary differential

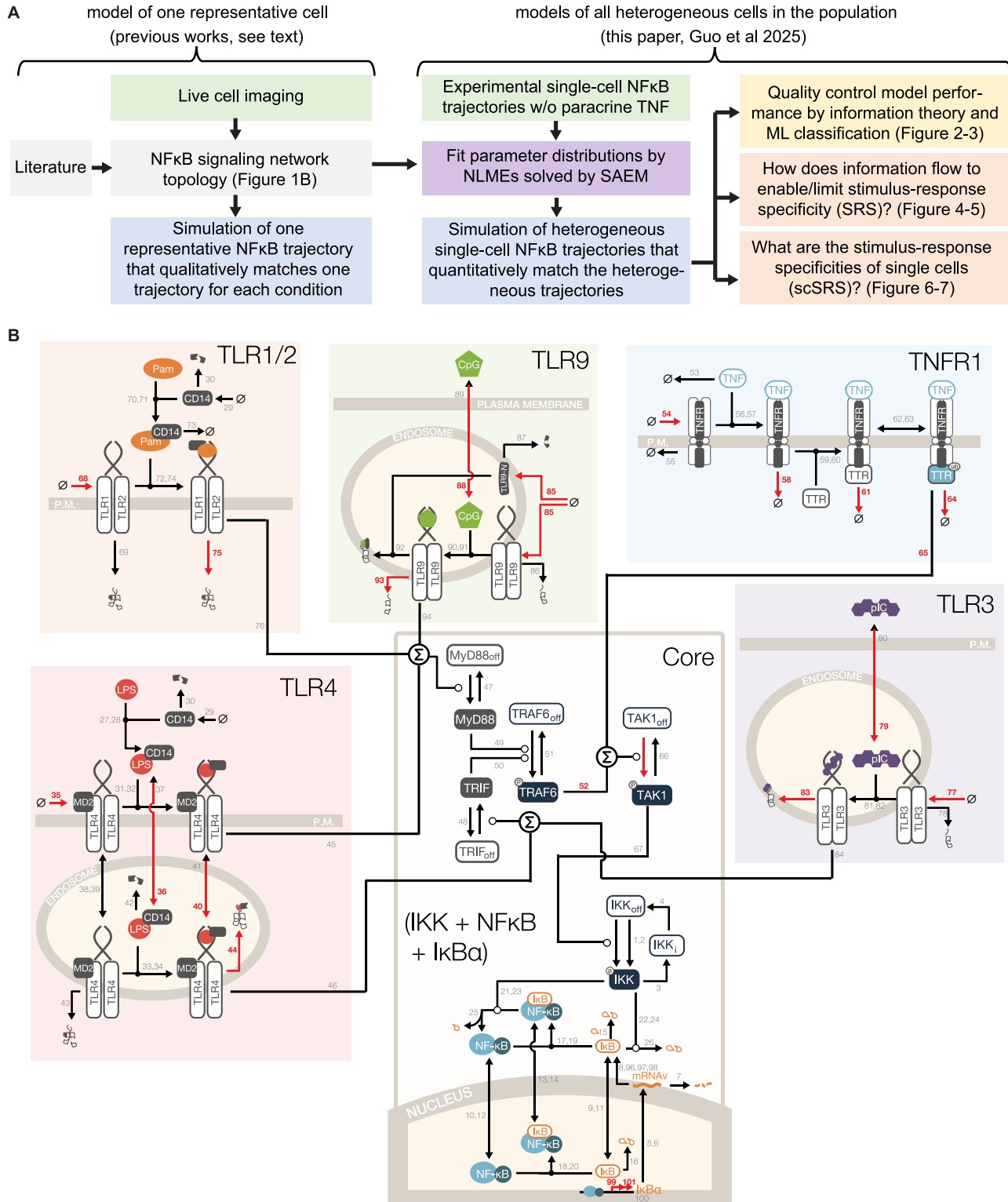

**Fig. 1 | Mathematically modeling heterogeneous single-cell signaling networks. A** Schematic of the goals and scope of the present study and how it builds on and advances prior work. While prior work established the signaling network topology that may account for the stimulus-response dynamics of a single cell representative of the population, the present work aims to identify parameter distributions that account for the stimulus-responses of all single cells in the population, via Non-Linear Mixed Effects model (NLME) and Stochastic Approximation Expectation-Maximization (SAEM) algorithm. Such model simulations account for the heterogeneity of the stimulus responses and enable systematic quantitative studies of information transmission by the population and of single cells. **B** Schematic of NFκB signaling network model, consisting of 5 receptor modules (distinguished by different boxes) and the core module encompassing TAK1, IKK, and NFκB-IκBα feedback. Red arrows and numbers denote the variable parameters for estimation. The schematic was adapted from Adelaja et al.[21]: the box colors have been updated to enhance visualization; the arrows representing the parameters for model fitting are marked red; and the reaction indexes have been revised.

**Table 1 | The number of single-cell NFκB trajectories under each experimental condition as defined my ligand identity and dose**

| Ligand (unit) | TNF (ng/mL) | | | Pam (ng/mL) | | | LPS (ng/mL) | | | CpG (nM) | | | | | pIC (µg/mL) | | |
|---|---|---|---|---|---|---|---|---|---|---|---|---|---|---|---|---|---|
| Dose value | 0.1 | 1 | 10 | 10 | 100 | 1000 | 1 | 3 | 10 | 33 | 100 | 10 | 33 | 100 | 333 | 1000 | 10 | 33 | 100 |
| Dose symbol | L | M | H | L | M | H | L | M | H | | | | L | M | H | | L | M | H |
| Number of cells | 420 | 744 | 645 | 507 | 612 | 596 | 176 | 245 | 437 | 310 | 327 | 258 | 247 | 381 | 308 | 394 | 666 | 540 | 557 |

equations of the NFκB signaling network model defined the non-linear function in the NLME, and parameters were subject to both fixed effects and random effects. To infer a single parameter set for each cell, we first solved the maximize likelihood estimator (MLE) of the distributions of selected parameters in the NLME via the stochastic approximation of expectation-maximization (SAEM) algorithm[48–51] given the experimental data (see Methods -- Single-cell model fitting), due to its excellent performance on benchmarking NLME problems and the availability of well-developed software[52]. For each individual single cell, we applied the maximum a posteriori (MAP) to estimate the modes of parameter posterior distributions given each its NFκB trajectory and the estimated parameter distribution. To avoid overfitting, 17 parameters were selected, comprising 2 or 3 parameters from each receptor module and 4 parameters from the core module (Fig. S1B–H). The selection criteria informed by biological relevance and experimental studies (Methods – Parameter selection and sensitivity analysis), such as receptor synthesis[46,53,54], endosomal trafficking[35], and signaling complex degradation[55]. These parameters were previously identified as potential drivers of cell-to-cell heterogeneity[35,42].

## Model simulations recapitulate the stimulus-specificity of NFκB signaling dynamics

Fitting to the experimental dataset of stimulus responsive single-cell NFκB signaling trajectories (Table 1), we obtained model-simulated single-cell NFκB trajectories calculated from the individual parameter estimates (Methods – Single-cell model fitting). We first assessed the goodness of fit through visual comparison between heatmaps of experimental and simulated NFκB trajectories (Fig. 2A). The model simulation captures key features such as the total activity, as the exemplified by difference between medium-dose LPS vs medium-dose Pam, the speed of NFκB activation, as exemplified by high-dose LPS vs. high-dose pIC, or the oscillatory content, as exemplified by high-dose TNF vs high-dose LPS. Further, the heterogeneity within each condition is also recapitulated well such as the activation speed in the high-dose pIC condition or the duration in response to high-dose LPS. While the cell-to-cell heterogeneity of oscillatory dynamics is nicely represented in the simulations, the simulated data are smoother than the experimental data, which are also affected by measurement or intrinsic noise. We quantified root mean square deviation (RMSD) between the experimental and simulated NFκB trajectories for every cell (See Methods Model performance evaluation), revealing RMSD distributions that ranged from 0 to 0.06, with a median between 0.01 and 0.03 across the 15 stimulus conditions (Fig. S2A). Between 65% and 90% of the RMSDs for individual fitted trajectories under each condition are below the threshold of 0.03 (Fig. S2B).

To determine whether the model recapitulates the dynamic features that encode stimulus-specific information of NFκB signaling[21], we decomposed trajectories into signaling codons for both experimental and simulation data. The six signaling codons are activation speed (Speed), maximal value (Peak), duration time (Duration), integral or total activity (Total), early-vs-late activity (EvL), and oscillatory content (Osc) of the NFκB activation (see Supplementary Notes for details, Fig. 2B, upper panel). The signaling codons of experimental and simulated data for different stimulation conditions show similar distributions (Fig. S2C). We then calculated the average of Wasserstein distances, across the six signaling codon distributions of the same stimulus condition or different stimulus conditions (Methods – Model

performance evaluation). The Wasserstein distance measures the cost of transforming one distribution into another, resulting in larger values when the distributions are separated further. Most (83%) Wasserstein distances between experimental stimulus conditions were greater than 0.1, and 56% were greater than 0.15, similar to computational simulation data. In contrast, the distance between corresponding simulated and experimental stimulus conditions was generally (93%) less than 0.1, with 53% of the data being <0.05 (Fig. 2B, lower panel). Examining the average Wasserstein distances between corresponding experimental and simulation conditions (Fig. 2C, diagonal) confirmed smaller distances than those between differing stimulus conditions of the experimental (lower left half) and simulation (upper right half) data (Fig. 2C). In addition, there is symmetry in the pattern of Wasserstein distances in the experimental vs simulation halves of the distance matrix, confirming a good alignment between computational and experimental signaling trajectories. For example, the model simulations recapitulate the similarities among responses to bacterial PAMPs ligands (Pam, CpG, and LPS) and the specificity of cytokine (TNF) and viral PAMP (pIC) stimulated condition. This is evidenced by the small Wasserstein distances among bacterial PAMPs in both experimental (lower triangular blue matrix among conditions Pam, CpG, and LPS) and simulation (upper corresponding triangular blue matrix) data (Fig. 2C). Further examination of individual signaling codons identified that Total was generally best fit with lowest Wasserstein distance, but that other signaling codons also showed generally low values, with Oscillatory content showing low values in the majority of stimulus conditions (Fig. S2D). Overall, signaling codons of NFκB responses to five ligands across different dosages showed good agreement (Fig. S2E).

To quantify if the model simulation possesses the same SRS performance characteristics as live cell macrophages, we applied machine learning random forest classification using signaling codons as inputs and ligand information (at high doses) as outputs (Methods –Model performance evaluation). High-dose conditions were selected because they provide the highest SRS[21]. We found similar patterns of confusion between the simulation and the experimental data (Fig. 2D, Fig. S2F, G). Bacterial PAMPs were more likely to be confused with one another, whereas the cytokine TNF condition was the most distinguishable, followed by viral PAMPs, which were also well distinguished. In summary, quantitative fit assessment via signaling codon decomposition and SRS performance confirms that the model simulations capture the stimulus-specific responses that are hallmarks of macrophage NFκB signaling. An in-depth analysis of the parameterized model revealed that signaling codons are co-determined by multiple single-cell biochemical rate constants *via* highly non-linear, non-monotonic relationships (Supplementary Notes), consistent with a complex, feedback-containing dynamical system.

To assess the robustness of the model results across different sample sizes, we down-sampled the Pam condition dataset to half its original size and evaluated the model's performance. The down-sampled dataset maintained an acceptable goodness-of-fit, exhibiting similar NFκB trajectories between experiments and simulation (Fig. S2H), comparable signaling codon distributions (Fig. S2J–I), and minimal Wasserstein distances between experimental and simulated data while recapitulating dose differences (Fig. S2K, L). In contrast, when fewer parameters were included in the estimation the fit quality suffered (Fig. S2M).

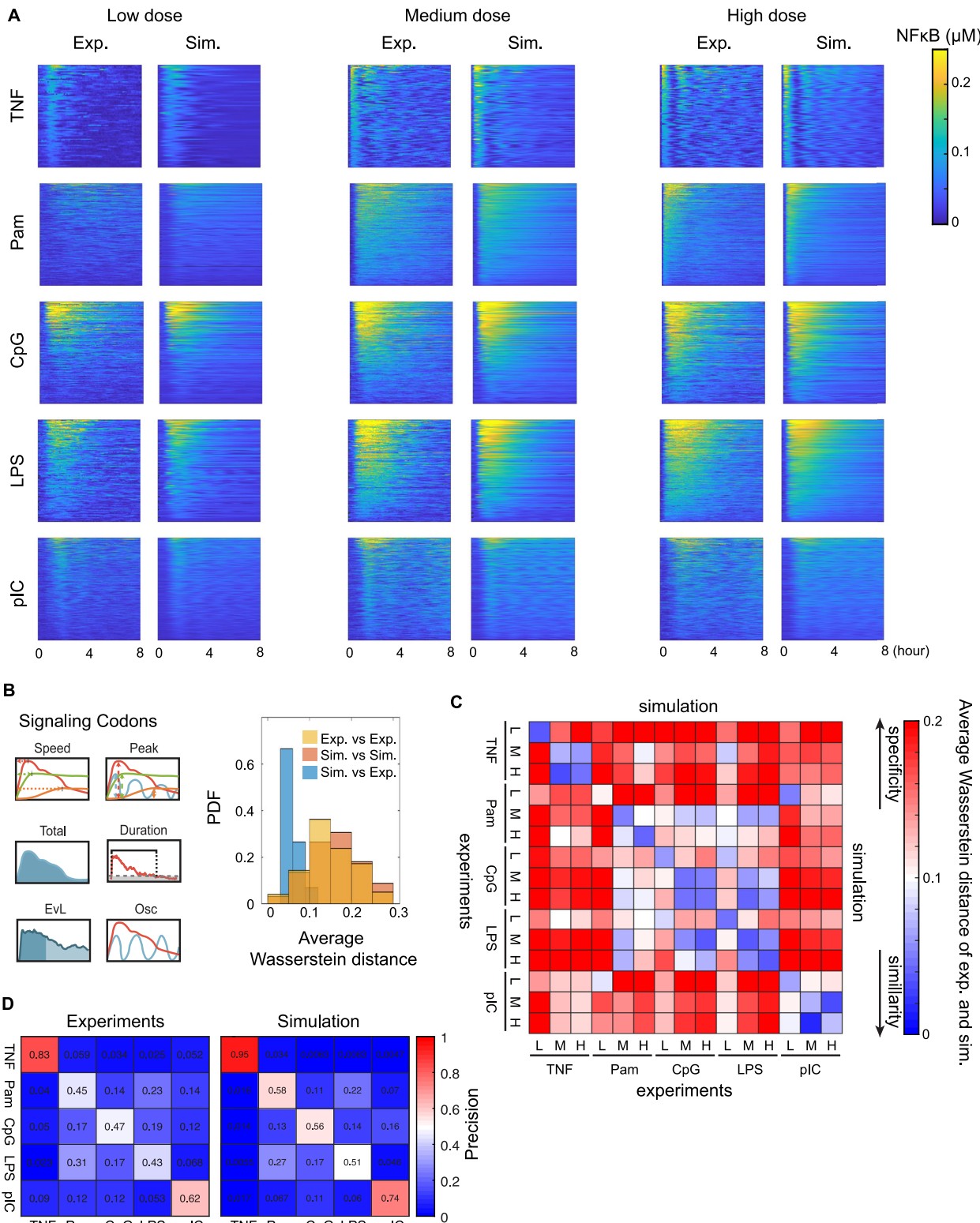

**Extended dose response studies reveal true limits in channel capacity**

We next explored the parameterized model's utility as a research tool for generating new datasets in silico to complement and extend experimental studies and address questions that are not easily addressed experimentally. We developed a workflow in which parameter sets are sampled from fitted parameter distributions (Methods – Generating new dataset of NFκB trajectories). The reliability of this sampling approach was confirmed by first simulating stimulus conditions obtained in experiments, showing good agreement such as in oscillatory patterns, heterogeneity in total activity, etc (Fig. S3A). Decomposing the trajectories into signaling codons confirmed similar distributions (Fig. S3B) with an average Wasserstein distance around 0.1 or less (Fig. S3C). Finally, stimulus-response specificity as ascertained by machine learning classification showed similar patterns of precision and confusion scores as the original model and the experimental data with only minor deviations (Fig. S3D).

**Fig. 2 | Comparing experimental data and model simulations of NFκB signaling in macrophages. A** Heatmap displaying both experimental (left columns) and simulated (right columns) NFκB signaling in macrophages stimulated by indicated ligands (rows: TNF, CpG, Pam, LPS, pIC) and doses (columns: low, medium, high). For each condition, the experimental data (left, exp.) and model simulations (right, sim.) are shown. In each subpanel, y-axis represent single cells, the x-axis indicates time, the color intensity signifies NFκB abundance according to the color bar. Experiments used here were a representative of ten replicates, some of them were published in previous studies[5,21,23,24]. **B** Dynamic features of signaling trajectories (left), that had been identified as being informative of ligand and dose and termed Signaling Codons[21] were used to decompose the NFκB signaling trajectories from experiments and simulations. Histogram of average Wasserstein 2-distance (a distance function defined between probability distributions on a given metric space, the Euclidean space in our case) between distributions of Signaling Codons from experiments and simulations (right), with distances between different experimental conditions in yellow, between different simulation conditions in orange,

and distances between model simulations and experimental data in blue. **C** Heatmap depicting the average Wasserstein distances between Signaling Codon distributions within and between experiments and simulations across different conditions. The diagonal elements (mostly blue) represent distances between corresponding experimental and simulation data for each condition. The lower triangular matrix illustrates the distances among different experimental conditions, while the upper triangular matrix shows the distances among various simulation conditions. Conditions are specified along both the x- and y-axes (L: low dose; M: medium dose; H: high dose). The color intensity indicates the magnitude of the average Wasserstein distance. **D** Confusion matrices illustrating the classification precision of ligand identity information for experimental data (Experiments), and for simulated data calculated and from fitted parameters (Simulation). The machine learning model (Random Forest) uses NFκB trajectory Signaling Codons as input to encode and predict the ligand information. This is evaluated by five-fold cross-validation.

To validate the model's forward prediction for extended doses, we compared its predictions beyond the training range with published experimental data for 33 ng/mL TNF and 333 ng/mL LPS[21]. The model predictions align well with experimental NFκB trajectories (Fig. S3E) and effectively recapitulate Speed, Peak, Total, and Osc signaling codon distributions, and to lesser extent Duration and EvL (Fig. S3F). The average Wasserstein distances between model-predicted and experimental signaling codon distributions were less than the differences between ligands (Fig. S3G).

A fundamental question in signal transduction is how well signaling pathways are able to discriminate ligand doses, prompting the recent application of information theoretic approaches[25,47]. Experimental studies are necessarily limited to a finite number of doses, and in our current experimental dataset we used three doses for each stimulus informed by prior studies[21]. Here, we used the our established mathematical model to extend the dose-response studies. NFκB response trajectories were simulated for 20 doses of each ligand, ranging from an almost negligible to a supersaturating amount, spanning five orders of magnitude. The model recapitulates that with increasing dose, the proportion of activated cells first rises, exhibiting single-cell heterogeneous activation, and then the response saturates where neither responder fraction nor intensity increase further (Fig. 3A).

To determine how much stimulus-dose information is contained in the NFκB trajectories we calculated the maximum mutual information (MI) between dose stimulation and the resulting NFκB signaling codon distributions (Fig. S3H) (See Methods Maximum mutual information calculation), using either experimental or simulated data (Fig. 3B). For the ligands Pam, CpG, LPS, and pIC, the information content of the chosen experimental dose was similar (about 1 to 1.2 bit respectively). The extended set of 20 simulated dose responses did not result in higher maximum MI in the case of Pam, CpG and pIC, but did in the case of LPS. Indeed, previous experimental studies with five or more doses of LPS resulted in 1.3 bits of information[21,25]. Interestingly, in the case of TNF, the maximum MI in simulated 3-dose NFκB trajectories was higher than in the experimental data, and a larger number of doses further increased the maximum MI values. In simulations of TNF condition, without technical and intrinsic noise, the EvL codon for low doses (red distributions in Fig. S3I) can be distinguished from the unstimulated condition (gray), and medium-dose (green) from high-dose (blue) conditions, as high-dose cells show more sustained signals with higher duration and later activity (small EvL codon) (Fig. S3I). However, in experimental TNF data, the low dose condition are not distinguishable from unstimulated condition (Fig. S3J), as the peak detected in low dose is similar as noise observed in unstimulated conditions. Medium- and high-dose TNF conditions are also indistinguishable (Fig. S3I). This divergence stems from Duration and EvL which show higher distinction between doses in the simulated than the

experimental data (Fig. S3I, J). These two metrics might be more sensitive to technical noise when trajectories are oscillatory.

## Locating information loss: the receptor module

Given that NFκB dynamic trajectories do not show perfect stimulus distinction (Fig. 2D), we asked where information loss occurs within the signaling pathway that generates NFκB trajectories. Unlike experimental models, a mathematical model allows us to manipulate and assess the impact of noise in different modules of the network. Here, we distinguished the receptor module, adapter module, and IKK-IκB-NFκB core module (Fig. 4A), and eliminated noise (collapsing the parameter distribution to a single value) from one or more of these modules to assess each module's capacity for faithful signal transduction (Fig. 4B, Methods – Modeling different genotype, Denoise different modules of NFκB signaling network). We calculated maximum MI between the input stimuli (ligands) and the output NFκB signaling codons in response to five different ligand stimulations, resulting in 1.2 bits. Denoising the receptor module - collapsing the associated parameter distributions to a single value - increased the maximum MI by 0.9 bits to 2.1 bits, whereas denoising adaptor module or core module had little effect (Fig. 4C). We tested different single values and found the conclusions largely unaffected (Figure S4A). To complement the denoising approach, which tests for requirement, we asked whether noise in the receptor module is sufficient to degrade information transmission. To implement this, we started with the representative cell[21], which shows perfect stimulus distinction, and allowed for parameter variation according to the fitted distributions) only in one of the three modules (Fig. 4D). Adding extrinsic noise to either the adaptor or the core module had little impact on the mutual information, but implementing distributed parameters to the receptor module degraded the mutual information from >2.3 to 1.4.

Schematizing the information flow through the NFκB signaling pathway based on the above results (Fig. 4E), we may say that of the 2.32 bits of information associated with 5 ligands (theoretical maximum), about 0.9 bits are lost in the ligand-proximal receptor modules, and only about 0.2 bits are lost in the shared adaptor module and almost none in core module. These conclusions align with previous experimental findings indicating that upstream signaling modules can be the information bottleneck[47] and that NEMO and RelA dynamics are highly correlated[33]. To investigate whether information loss in the receptor module can be attributed to high parameter variation, we calculated the coefficient of variation (CV) for each parameter but found similarly levels of variation (Fig. 4F). This indicates that while parameters vary similarly the variation in the receptor module is primarily responsible for information loss.

Given the remarkable signaling fidelity of the IKK-IκB-NFκB core module, we asked whether the prominent negative feedback that IκBα provides is critical for this. We modeled an IκBα promoter mutant

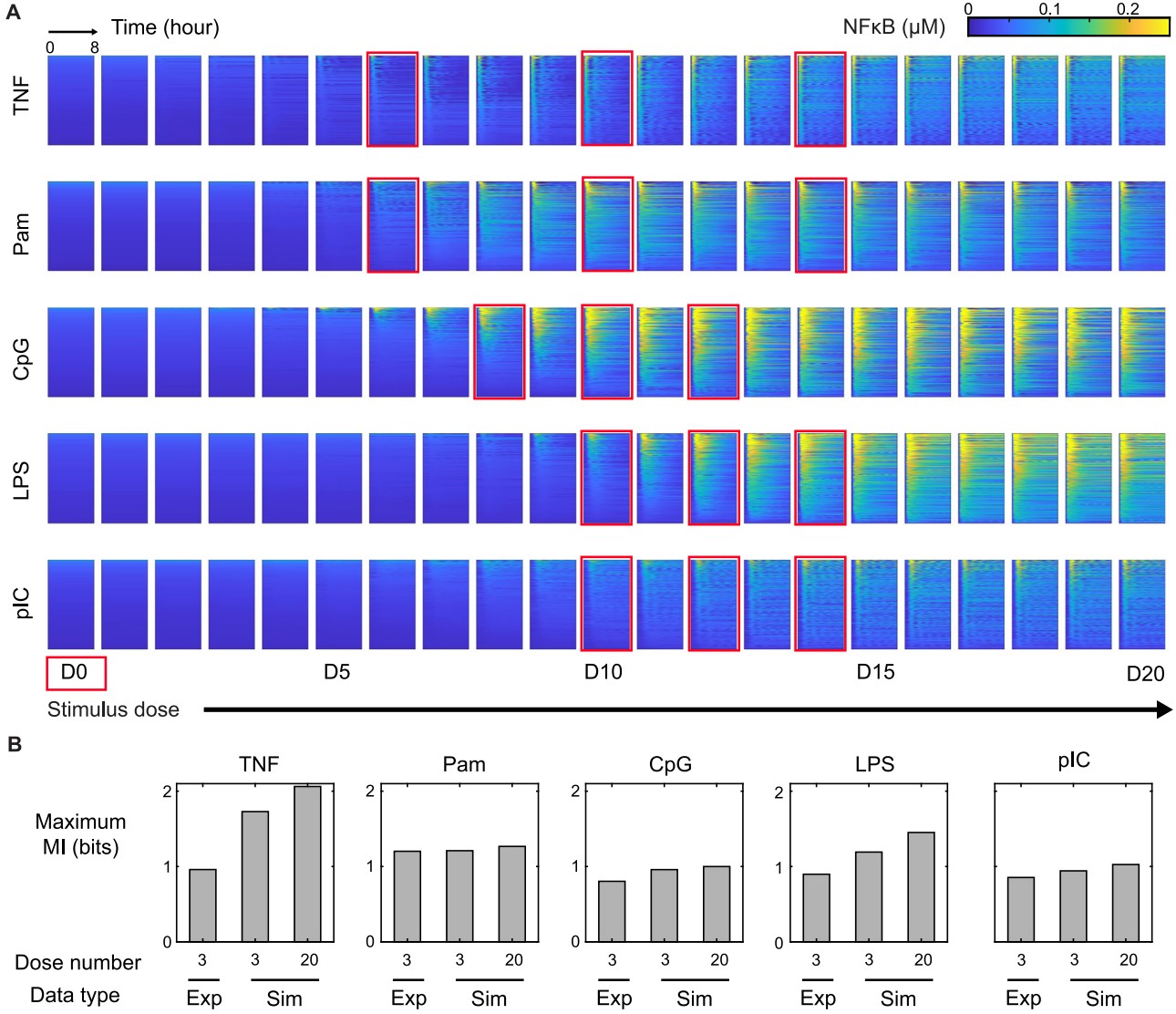

**Fig. 3 | Using model simulations to estimate the channel capacity for encoding stimulus doses. A** Heatmaps of simulated NFκB signaling trajectories in immune cells stimulated by 21 different doses (columns) of 5 ligands (rows), calculated from the sampled parameters. Within each heatmap, the x-axes denote time (hours), while y-axes represent different single cells ordered by Total signaling activity. For TNF and LPS, the 21 doses span log-linear points from $10^{-2.5}$ to $10^{2.5}$ ng/mL; for CpG, from $10^{-0.5}$ to $10^{4.5}$ nM; for pIC, from $10^{-1.5}$ to $10^{3.5}$ μg/mL; and for Pam, from $10^{-0.5}$ to $10^{4.5}$ ng/mL. D0 is the lowest dose for each ligand, while D5, D10, D15, and D20 correspond to doses that are 5, 10, 15 or 20 increments higher. Red squares indicate the approximate doses that were used for experiments. **B** Channel capacity analysis between ligand and signaling codons, either for the 3 doses or all 20 with unstimulated condition using experimental (exp) or sampled simulated (sim) NFκB trajectories. The 3 doses of exp and sim is indicated in Table 1 and in Fig. 3A red box, respectively, where the 3 doses of sim imitate experimental doses.

(IκBα^S/S) by reducing the NFκB-mediated transcription rate of IκBα by 75% (Fig. 5A)[23] (Methods – Modeling different genotype, IκBα promoter mutant). The model predicts reduced oscillations, extended response duration, and increased total activity in IκBαS/S, while signaling codon, Speed, and EvL remain comparable to WT (Fig. S5A, B, top rows), and analysis of published data[21] confirms the predicted reduction in oscillations with other codons unchanged (Fig. S5A, B, bottom rows). To evaluate whether the model predictions exhibit similar stimulus-response specificity (SRS) performance as live macrophages, we applied a machine learning random forest classifier to model-predicted NFκB signaling codons for WT and IκBα^S/S and compared to the experimental dataset. The model predictions revealed increased confusion patterns for the IκBα^S/S compared to WT, aligning with experimental data (Fig. 5B). These results suggest that the IκBα negative feedback is not only important for terminating NFκB activity or generating oscillations[29], but is also a means for faithful information transmission by minimizing information loss. These studies are yet

another example of the power of a mechanistic model, as knowledge of the chemical reaction network structure is required to locate where information loss occurs.

**Model reveals the stimulus-response specificities of single cells**
The literature of quantifying SRS in cellular signaling has focused on the population level, either via MI between signaling feature distributions and stimuli, or machine learning classification[21,23,24,56] (Fig. 6A, left panel). The population focus is required because experimentally, a single cell can only be stimulated with a single stimulus condition – in a sequential stimulation regime, the response to the second stimulus is affected by the response to the first[57]. However, simulating a mathematical model of a signaling network may be used for obtaining the signaling responses to different stimulus conditions (Fig. 6A, right panel). Here, we generated single-cell NFκB signaling networks for multiple single stimulus-response simulations (Fig. 6A right panel). The resulting set of stimulus-response trajectories associated with

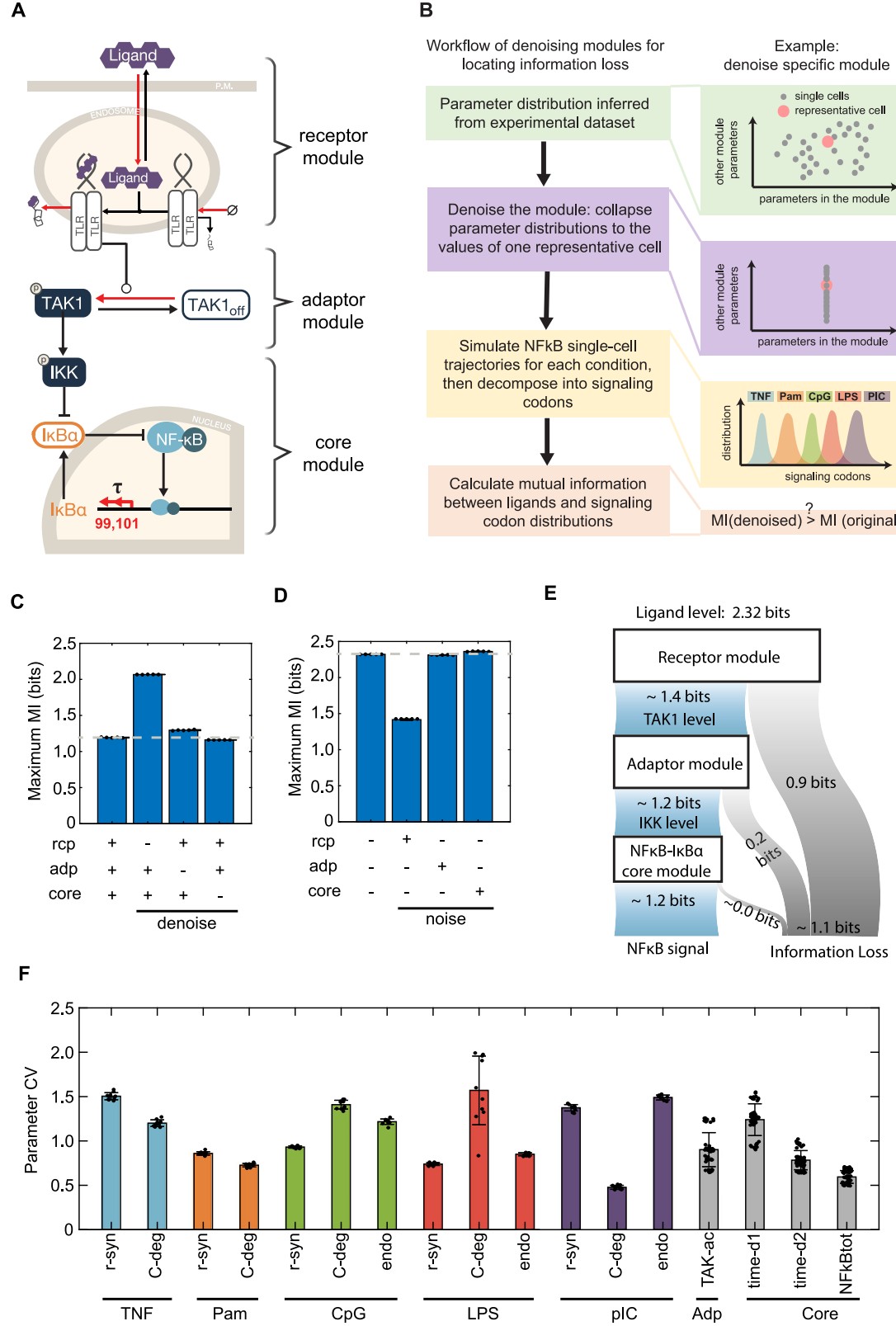

each single-cell NFκB signaling network is then assessed for single-cell SRS (scSRS) by calculating the differences in NFκB signaling codons. This workflow reveals how individual single-cell NFκB signaling networks differ in their capacity to produce stimulus-specific NFκB responses.

Using the collection of single-cell NFκB signaling networks produced by combining inferred parameters (Methods – Generating new

dataset of NFκB trajectories), we generated single-cell stimulus response data for five ligands at high doses. Ordering the data for all five stimuli based on the TNF response integral revealed that a high response integral to TNF does not necessarily correlate with a high response integral to other ligands (Fig. S6A). Hierarchically clustering the stimulus-responses of a subset of single cell NFκB signaling networks illustrated this more generally: a high response to one ligand

**Fig. 4 | Information flow within the NFκB signaling pathway. A** Schematic of the NFκB signaling network. Extrinsic noise is represented by parameter variation in the Receptor, Adaptor, and Core modules. **B** Workflow for identifying information loss by denoising individual signaling modules. This involves collapsing the distribution of the respective parameters to a single value. **C** Maximum mutual information (Maximum MI) between the ligands and NFκB signaling codons for the original network and after denoising the Receptor (rcp), Adaptor (adp), or Core (core) modules (labeled at the bottom). MI is calculated five times per condition, shown as individual data points. Data are presented as mean values ±SD. **D** Maximum MI of the network under no noise, noise added to the Receptor module (rcp), Adaptor module (adp), or core NFκB-IκBα module (core) (labeled at the bottom). The no-noise condition is based on calculations from 1000 identical cells. MI is calculated five times per condition, shown as individual data points and as mean values ±SD. **E** Illustration of the successive information loss along the pathway due to noise in

indicated modules. Blue shade represents information flow, gray shade represents information loss. **F** Bar plots of average coefficient of variation (CV) of Receptor (TNF, LPS, CpG, pIC, and Pam), Adaptor (adp), and Core (core) module parameters, calculated from 10 permutation-sampled datasets, shown as data points and as mean values ±SD. 'r-syn' is the receptor synthesis rates (k54 for TNFR, k68 for TLR1/2, k85 for TLR9, k35 for TLR4, k77 for TLR3); 'C-deg' is the signaling complex degradation rate (k58, k61, and k64 for TNF module, k75 for Pam module, k44 for LPS module, k83 for pIC module); 'endo' is the endosomal import rate (k88 for CpG module, k36 and k40 for LPS module, k79 for pIC module); 'TAK-ac is the TAK1 activation rate (k52 and k65); 'time-d1' and 'time-d2' is the time delay parameters in NFκB mediated transcription of IκBα (k99 and k101); NFκBtot is the total initial NFκB abundance within single cell. Colored bars indicate receptor modules, gray bars indicate adaptor and core modules.

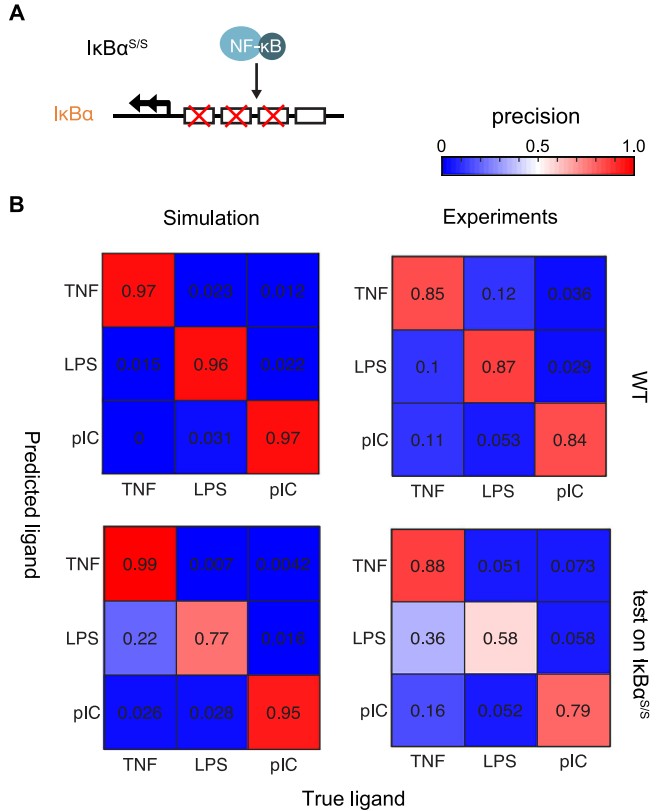

**Fig. 5 | Model prediction and testing of NFκB responses in Sjögren macrophages. A** Schematic of the IκBα$^{S/S}$ promoter mutation which underlies the Sjögren macrophages studied previously[21]. To mathematically model this mutation, the NFκB-mediated transcription rate of IκBα was reduced four-fold. **B** Confusion matrices illustrating the classification precision of ligand identity information for experimental data (Experiments)[21], and for simulated data calculated and from fitted parameters (Simulation), of the wild type (WT) and tested on IκBα$^{S/S}$ network. The machine learning model (Random Forest) used NFκB Signaling Codons as input to encode and predict the ligand information. The classifier was trained on the WT dataset, and then tested on WT and IκBα$^{S/S}$ datasets.

stimulus pairs each of the 1000 single cell signaling network models can distinguish. We found that 36.7% of single cells are capable of distinguishing all 10 stimulus pairs, and another 39.3% can distinguish 9 of the 10. Only 11.8% of single cell networks confuse 3 or more pairs (Fig. 6E). We found that the most frequent confusion occurs between LPS vs. Pam (18.6%), and the least confusion frequency is between TNF vs. CpG (2.8%) (Fig. 6F). Generally, confusion between TNF vs. PAMPs was less frequent (2.8–8.9%) than confusion among bacterial PAMPs (15.6–18.6%), with viral vs. bacterial ligands being in between (6.8 and 18.0%). These conclusions are robust across different specificity thresholds from 0.5 to 1.5 (Fig. S6F).

To investigate the role of the IκBα feedback in scSRS, we simulated single-cell responses of the IκBα$^{S/S}$ promoter mutant cells (Fig. 5A, S6C). Hierarchical clustering of single-cell stimulus-responses showed apparently diminished specificity in stimulus responses (Fig. S6D). By calculating the L2 differences in the signaling codon space for pairwise stimuli, we found that IκBα$^{S/S}$ cells are more likely to have lower scSRS than WT counterparts (Fig. S6E). The 50th percentile of the L2 distance distributions were diminished for all stimulus pairs; 90th percentiles were lower for 9 pairs; the 10th percentiles were lower for 6 of 10 pairs. The IκBα$^{S/S}$ genotype reduced the proportion of cells being able to distinguish all stimulus pairs by 5% (of 1000 cells) compared to WT (Fig. 6E, G). The L2 differences revealed increased confusion between TNF vs. PAMPs (7.6% of 1000 cells), among bacterial PAMPs (18.3% of 1000 cells), and between bacterial and viral PAMPs (1.7% of 1000 cells) (Fig. 6F, H). Among the cells confusing multiple stimulus pairs in IκBα$^{S/S}$, defective distinction occurs concurrently among bacterial PAMPs, similar as WT. Such confusion between cytokine and PAMPs may be physiologically particularly consequential[5,6,21].

## Small consortia of macrophages may achieve perfect stimulus response specificity

Given the inferred collection of single-cell NFκB signaling networks, we asked whether their heterogeneous stimulus-responses are subject to correlations that may result in patterns in which stimulus-pairs are confused. To this end, we mapped the instances of stimulus-pair confusion for each of the 1000 cells generated by the model for the specificity threshold of 1.0 (Fig. 7A). Consistent with Fig. 6E, we found that 36.7% of the single-cell NFκB signaling networks showed no confusion, and that 39.3% showed a confusion of a single stimulus pair. However, which stimulus pair is confused was highly diverse with no pair being distinguished by all cells. Further, in cells that showed confusion of two or more stimulus pairs, the instances of confusion were uncorrelated, such that the Jaccard distances between stimulus pairs are uniformly high, with stimulus pairs involving Pam or pIC showing a higher propensity for confusion (Fig. S7A).

This observation led us to the hypothesis that uncorrelated macrophage stimulus-responses that result in highly diverse stimulus-pair confusion patterns may allow neighboring cells

does not necessarily predict a strong response to another (Fig. 6B). Indeed, we found only weak correlations of signaling codons deployed in response to each stimulus (Fig. S6B).

To assess the scSRS, we calculated the L2 distance between the signaling codon vectors of each stimulus pair for the collection of single-cell signaling networks, yielding distributions of L2 distances (Fig. 6C). Examining these, we found that TNF vs. PAMPs tended to be more specific that pairs of bacterial PAMPs. Considering a threshold of 1.0 which appears to demarcate stimulus pairs that are distinct vs. those that are barely distinguishable (Fig. 6D), we asked how many

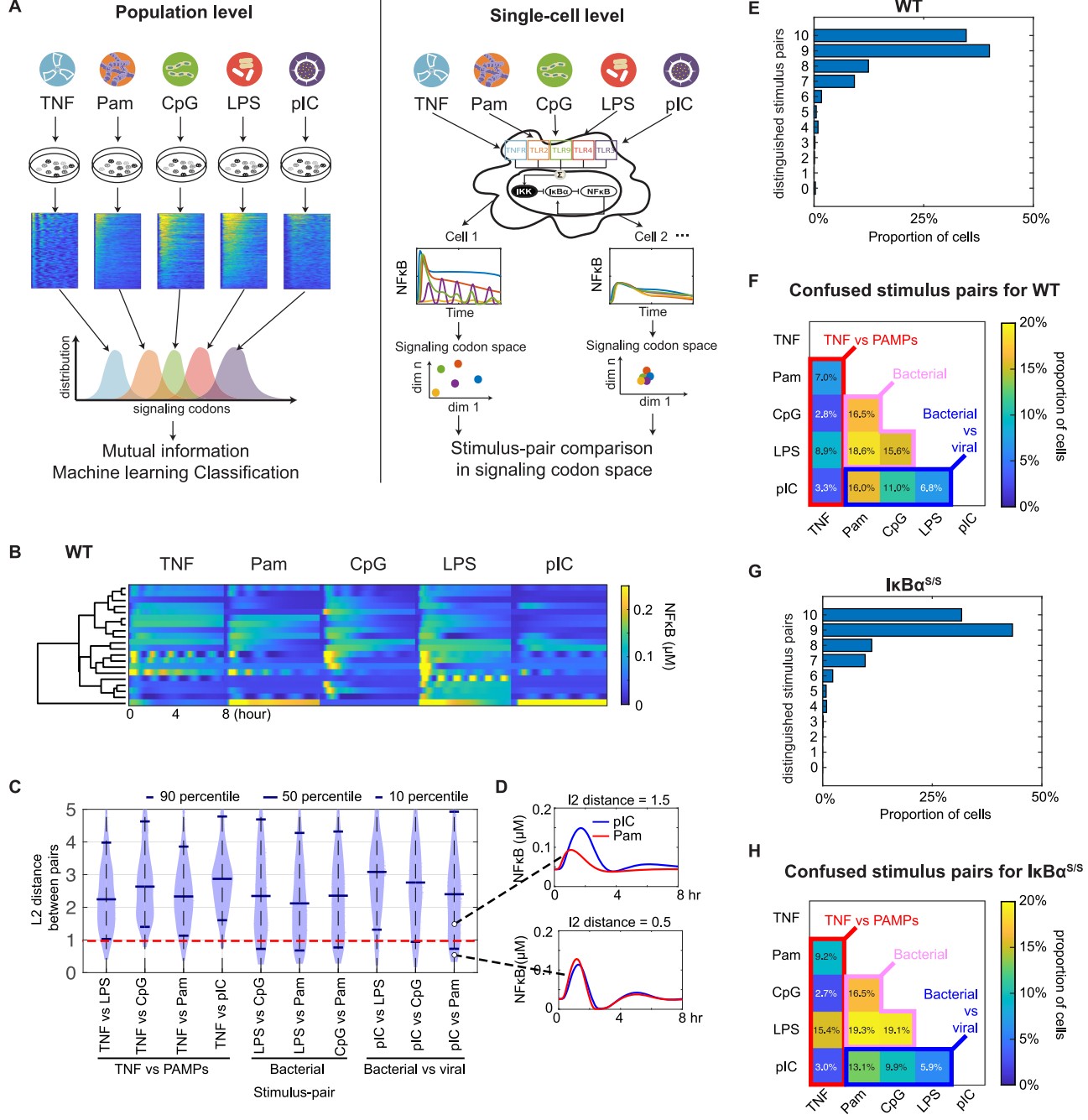

**Fig. 6 | Model prediction of single-cell stimulus response specificity (scSRS).**
**A** Schematics of how stimulus-response specificity (SRS) is assessed at the level of populations using experimental data (left), and at the single-cell level (right) as we propose using the mathematical model. Experimentally, macrophages are stimulated with different ligands in dedicated cultures, resulting in population-level distributions for each stimulus-condition of single-cell signaling trajectories or derived dynamic trajectory features. These distributions are then correlated by information theory or machine learning to the stimulus-identity. The proposed single-cell SRS (scSRS) is assessed by simulating the same mathematical model with its cell-specific parameter set with each of the ligands and then assess the pairwise differences in stimulus-responsive trajectories or derived dynamic trajectory features. This approach allows categorization of virtual cells as being able to distinguish stimulus pairs or being confused. **B** Heatmaps show simulated stimulus-responsive NFκB signaling trajectories in response to 5 ligands for 20 wild-type (WT) individual NFκB signaling networks (virtual cells). Each heatmap's x-axis

represents time in hours, and the y-axis lists individual virtual cells, ordered via hierarchical clustering. **C** Violin plots depicting the distribution of l2 distance in the signaling codon space between stimulus pairs (specified in the x-axis) for wild type. 10th, 50th, and 90th percentiles are marked from bottom to top within each violin plot. Red dashed line indicates the threshold at which the stimulus pair is considered distinguished or confused. **D** Representative single-cell trajectories of distinguished (upper panel) and confused (lower panel) stimulus pairs (Pam: blue, pIC: red), with L2 distances of 1.5 (>1) and 0.5 (≤1), respectively. **E** Histogram of proportions of individual NFκB signaling networks for wild type (WT) that are able to distinguish the indicated number of stimulus-pairs. **F** Heatmap of the proportion of individual NFκB signaling networks that confuse the indicated stimulus pairs (specified on x-axis and y-axis). The red box highlights TNF-PAMPs confusion, the pink box indicates bacterial PAMPs confusion, and the blue box represents bacterial-viral PAMPs confusion. G. Histogram analogous to (G) for IκBα^S/S. H. Heatmap analogous to (F) for IκBα^S/S.

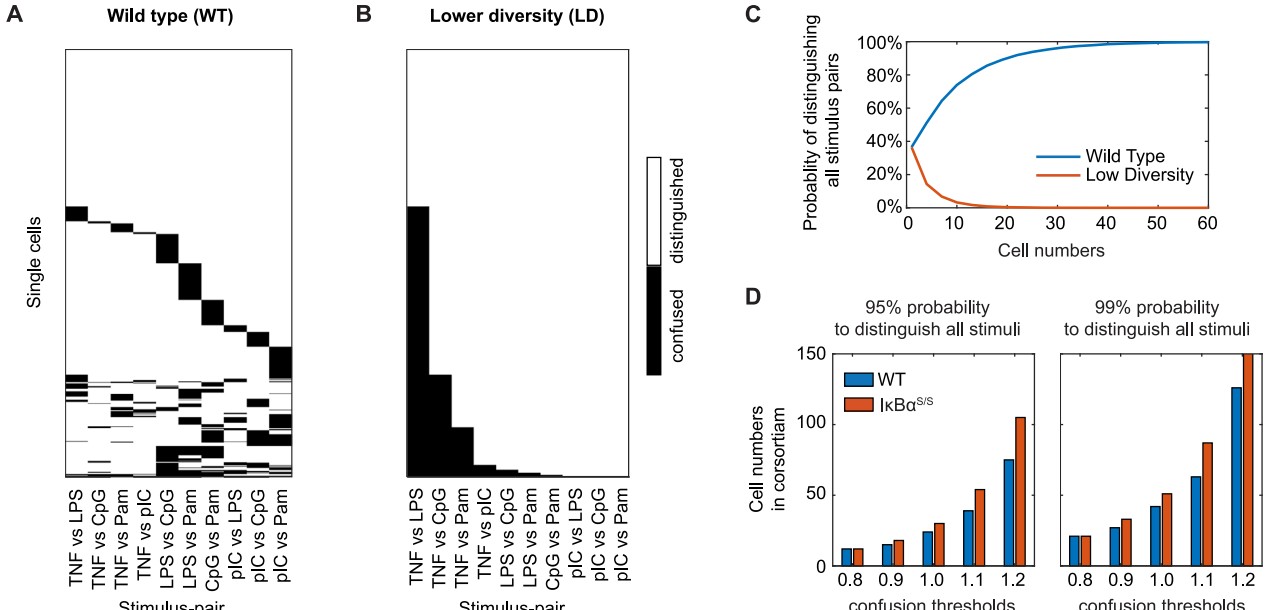

**Fig. 7 | Heterogeneity in scSRS enables small consortia of macrophages to achieve perfect stimulus response specificity. A** Binary category map indicating confused stimulus pairs (specified on x-axis) of each individual NFκB signaling network (virtual cells) for threshold 1.0 (Fig. 6C). Analogous maps for other thresholds are shown in Fig. S7. These show strikingly uncorrelated patterns of stimulus-pair confusion. **B** Binary category map indicating confused stimulus pairs (specified on x-axis) for a synthetic population of single cell NFκB networks in which the stimulus-responses are equally heterogeneous but the instances of stimulus-pair confusion are correlated resulting in a population of cells of lower

diversity (LD). **C** Line graph of the probability of distinguishing all 10 stimulus pairs (defined by at least two-thirds of the cells in the consortium distinguishing each pair) as a function of the number of macrophages in a consortium, comparing cell populations in which the stimulus-pair confusion pattern has lower diversity (LD, orange line) than wild-type (WT, blue line). **D** Barplots of cell numbers required for the consortia to achieve a 95% (left panel) or 99% (right panel) probability of distinguishing all 10 stimulus pairs under different distinction thresholds (x-axis), for cell populations in which the stimulus-pair confusion pattern is wild-type (WT, blue bars) or IκBα^{S/S} (orange bars).

compensate for each other's blind-spots, thereby resulting in an effectively enhanced stimulus-response specificity (SRS) of the consortium. The rules that govern the consortium's decision making are of course key for this result. To test our hypothesis, we considered that the consortium is governed by a 2/3 majority rule, and we generated a synthetic dataset of cells with the same scSRS distribution of the population (Fig. 6E) but in which the pattern of instances of stimulus-pair confusion show low diversity (LD) (Fig. 7B). We then randomly sampled one, two, three, or more cells from the wildtype (WT) and LD populations to determine the probability that these small consortia of cells achieve perfect stimulus specificity, i.e., distinguish all stimulus pairs (Fig. 7C). As expected, sampling just one cell, the probability is the same for both populations, as 36.7% of cells perform with perfect stimulus specificity. However, when considering the small consortia of macrophages, WT populations outperform LD populations in the probability of achieving perfect stimulus specificity. In WT consortia the probability of achieving perfect stimulus specificity increases with the member number (90% for 20 cells), while it decreases in LD consortia (almost 0% for 20 LD cells). Thus, WT consortia can generate perfect SRS with a dozen or so cells. We considered higher or lower stringencies for defining stimulus-pair distinction (Fig. S7B). The probability of achieving perfect distinction in WT consortia increases with the number of cells and consistently outperforms that of LD consortia, regardless of the chosen stringency threshold (Fig. S7C). In sum, our analysis shows that diversity among single cells in which stimulus pairs they confuse enables macrophages to complement each other such that small consortia of macrophages can achieve perfect SRS.

We quantified the minimum number of consortia members required for a 95% or 99% probability of achieving perfect SRS for both WT and IκBα^{S/S} genotypes. For the WT consortia, a 95% probability of perfect SRS was reached with ~10–110 cells, and a 99%

probability with about 20–130 cells, depending on the specificity threshold (Fig. 7D). In contrast, IκBα^{S/S} consortia required additional 5–10 cells at the default threshold (=1) to achieve the same 95% or 99% probabilities. Even when applying different thresholds, IκBα^{S/S} consortia consistently required at least as many, and often more, cells than their WT counterparts. This suggests that although the IκBα^{S/S} genotype loses scSRS (Fig. 6G, H), the retained diversity within its single-cell responses still enables small consortia to effectively differentiate stimuli. However, it remains less capable than the WT genotype at achieving perfect stimulus distinction.

## Discussion

In this study, we developed virtual single-cell NFκB signaling networks that recapitulate the experimentally determined stimulus-response specificity (SRS) of heterogeneous NFκB responses present within macrophage populations. These mathematical models, unlike machine learning classifiers, encode a wealth of biological knowledge of molecular mechanisms enabling simulations in conditions that are not within the range of the training data, thereby leading to insights about how SRS of NFκB signaling is regulated, and revealing how the population level SRS, which is quantifiable from experimental data, is composed of a diversity of single-cell stimulus-response specificities (scSRS) of individual macrophages. Our analysis locates primary noise sources to receptor modules that results in imperfections in stimulus-distinction of high diversity that may therefore be compensated for within small consortia of macrophages.

To generate the virtual single-cell NFκB signaling networks we leveraged an established molecular network model that recapitulates the signaling dynamics of a representative cell[21], and parameterized it to newly generated single cell experimental data. Prior studies of parameterizing distributions of biochemical parameters involved reaction networks with less than 10 dimensions[50,51]. Given our 52-

dimensional highly nonlinear dynamical system with feedback, we selected a subset of biochemical constants for parameterization based on prior studies[21,35,42,57], their sensitivity in determining signaling codons (Fig. S1), while ensuring that analogous parameters were represented in parallel receptor modules to enable insights based on comparisons. We used a nonlinear mix-effects model for the estimation of parameter distributions. RMSD of trajectories and Wasserstein distance of signaling codon distributions revealed simulations well fitted to the experimental data. However, some imperfections were noted, particularly in the oscillatory NFκB response to TNF, which were more pronounced but more heterogeneous in the intra-trajectory oscillatory period in experiments (discussed below). Additionally, it is possible that other negative feedback regulators such as IκBε and A20 have an effect in macrophages and that compensatory parameter values in our model make up for not including them explicitly.

We first employed the model to examine whether prior estimates of dose-response channel capacity were affected by limited experimental data. Such estimates were based on 3 to 5 doses and yielded estimates between 1 and 1.5 bits[21]. Using the model we generated simulation for 20 doses of each stimulus (Fig. 3), and found that for CpG, Pam, and pIC the estimates remained at 1.0, and for LPS at 1.3, suggesting that the channel capacity for stimulus dose transmission is indeed substantially limited. However, in the case of TNF, our simulation data suggested that the pathway has higher channel capacity than the 1.0 bit that was previously reported[21,25]. This may be the result of having data for more doses available or the simulations being devoid of technical or intrinsic noise. We note that the experimental TNF-induced oscillatory trajectories are more pronounced and contain more intra-trajectory variability in period than the simulation data. The latter may diminish information transmission via Duration and EvL signaling codons and is due to intrinsic noise within the IκBα feedback loop. As intrinsic noise in the negative feedback may expand the resonance frequency and hence the robustness of oscillations[58], it appears that the oscillatory NFκB response to TNF, which avoids nucleosome eviction and epigenetic reprogramming[18,59], has evolved to be robust even at the expense of the ability to distinguish different doses of the cytokine.

Based on the parameterized single-cell models, we developed a workflow of denoise different modules to locate the sources of extrinsic noise that limit the channel capacity for information transmission. We approached this question, which cannot be addressed experimentally, by collapsing fitted parameter distributions and assessing the channel capacity of the signaling network (Fig. 4). Employing this approach, we located almost all information loss to the receptor module, aligning with prior studies that implied that an information bottleneck at the receptor level[47]. On the other hand, the common IκB-NFκB core module showed high signaling fidelity despite introducing substantial heterogeneity (Fig. 4). This is also consistent with previous findings that IKK activity is highly correlated with NFκB, indicating little information loss for the IκB-NFκB core module[33]. In the experimental studies, such a denoising strategy is not available, so measuring two or more species to study information flow within the network is an alternative approach[33,47,60]. For the systems with only one protein measurement feasible, our modular denoising approach provides an appropriate and practical method to study the information flow. This is because the key intermediate signaling molecules can be inferred from the underlying signaling network structure and single-cell trajectory data.

Finally, we leveraged the model to decompose the population level SRS measures into the SRS characteristics of individual cells. By integrating regulatory modules that were parameterized with datasets derived from different stimulus-response experiments, we generated a collection of single-cell NFκB signaling networks (Fig. 6) – in essence virtual macrophages in regard to NFκB signaling. This enables studying SRS at the single-cell rather than population level, but required the

development of appropriate metrics. To gain detailed insights we opted for pairwise comparisons rather than a composite measure. We found that >75% cells are capable of distinguishing either all 10 or 9 out of 10 stimulus pairs by generating stimulus-specific NFκB dynamics. And almost no cells are defective in the distinction of half or more stimulus pairs. By drilling into the scSRS profiles we found that while bacterial PAMPs are more likely to be confused than other stimulus pairs, the imperfections in stimulus pair distinction were highly diverse and uncorrelated. No stimulus pair was distinguished by all cells, and Jaccard distance analysis between stimulus-pairs that are confused showed a lack of correlation. Direct experimental validation of this prediction on diverse scSRS is challenging with available technology. For example, isolating recently divided sibling cells to stimulate them with distinct ligands faces the technical challenge of isolating and moving individual cells without perturbing their cell state[61]. An alternative approach may be to learn how cell responses to a stimulus are affected by prior stimulation, for example by developing expanded models to account for adaptive changes. Then by tracking individuals during repeated stimulation regimes, the appropriate computational interpretation could approximate scSRS profiles for the tracked cells.

We examined the functional consequence of this remarkable diversity in single-cell stimulus response specificity characteristics by constructing an analogous population that showed less diversity and more correlation in their imperfections in stimulus distinction (Fig. 7). Sampling from these populations we found that small consortia of dozens of cells could have a 95–99% chance of perfect stimulus distinction unlike their less diverse counterparts. This implies, while each cell may have blindspots, their diversity enables immune response specificity in the collective. This conclusion may also depend on the rules governing functions of consortia of cells. We considered that if majority of cells see distinctions, then they can override minority cells that do not. Alternatively, cells that confuse stimuli could dilute the stimulus response specificity of the consortium if they retain a vote.

In summary, using a mechanistic mathematical model carefully parameterized to a wealth of experimental data allowed construction of a population single cell NFκB signaling networks, and thereby deconvoluting population measures of SRS into the collection of scSRSs. We found that the fidelity of single cells in mounting pathogen-appropriate responses is higher than what is evident from population level SRS measures of maximum mutual information or machine learning classification. Further, we found that we cannot assume that single cells of the population have the same sensing capabilities[27]. Instead, they are highly diverse, enabling even a small consortium of cells to reach perfect stimulus distinction. These insights support the emerging view that macrophage responses are highly stimulus-specific and that their stimulus-response specificity is a hallmark of their physiological function.

## Methods

Our research complies with all relevant ethical regulations. The UCLA Institutional Animal Care and Use Committee approved the protocol for animal research per guidance from the American Veterinary Medical Association.

### Experimental data generation

We used a C57-Bl6 mouse line that expresses mVenus-tagged RelA[21] (strain B6(SJL)-Relatm2.1Alex/J) to generate the experimental dataset. Mice were housed in a dedicated vivarium room in conditions of ambient room temperature of 20–25 °C and humidity of 40–50% with a light/dark cycle of 7am–7pm in accordance with Institutional Animal Care and Use Committee (IACUC) guidelines under approved protocols. Bone marrow cells isolated from the femurs of 8–12-week-old mice were cultured in L929-conditioned medium to differentiate them into bone marrow-derived macrophages (BMDMs)[21]. On day 4, BMDMs were detached, re-plated in an 8-well ibidi SlideTek chamber to at

15–20,000 cells/cm². The culture medium was replaced with fresh medium on day 6. These BMDMs were then stimulated with specified concentrations (Table 1) of TNF (TNF-alpha (aa 80–235) protein, 410-MT-010, R&D Systems), CpG (synthetic CpG ODN 1668, tlrl-1668, Invivogen), Pam (Pam3CSK4, tlrl-pms, Invivogen), LPS (L6529-1MG, Sigma-Aldrich), and pIC (polyinosine-polycytidylic acid (Poly(I:C)), tlrl-picw, Invivogen). To block paracrine feed-forward TNF signaling during PAMP responses, we co-treated the cells with 5 μg/mL of soluble TNF receptor II (Recombinant Mouse sTNFRII/TNFRSF1B, 426-R2-050, R&D Systems). We assigned a coded numbering system to samples to reduce awareness of sample identities during processing, thereby achieving a degree of blinding to group allocation. Nuclear NFκB activity in individual cells was monitored by time-lapse live-cell fluorescence microscopy, with images captured every 5 min over an 8-h period. All fluorescence intensity values were normalized to the background signal and baseline-corrected using the MACKtrack automated image analysis pipeline (https://github.com/Adewunmi91/MACKtrack). More details are provided in the Supplementary Notes.

## Computational model parameter selection and sensitivity analysis

We employed 52-dimensional ordinary differential equations (ODE) to model the NFκB signaling pathway, adapted from prior research[21], which comprises 52 variables (Supplementary Dataset 1), 101 reactions (see Supplementary Notes for equations), and 120 parameters (Supplementary Dataset 2). The single-cell parameters of the NFκB network ODEs were inferred based on experimental dataset to recapitulate the heterogeneity (explained in detail in next section). To avoid overfitting, only a limited number of parameters were selected for fitting based on following four principles: (1) parameters with direct or strong biological evidence indicating heterogeneous measurements were prioritized, such as receptor abundances, trafficking capacities; (2) across different receptor module, consistent parameters are chosen, such as receptor synthesis rate; (3) redundancy in parameter functions was avoided—for instance, while both receptor synthesis and degradation rates can regulate receptor abundance, only one of them was selected, the receptor synthesis rate, as they alter NFκB dynamics in a similar way; (4) core module parameters were selected based on their capacity of generating oscillation heterogeneity. All other parameters were fixed to the values from the literature[21]. Based on these principles, the following parameters were selected (16 independent parameters in total): core module parameters, including TAK1 activation (TAK act, k52, k65), the time delay for NFκB-regulated IκBα transcription (time delay related rates, k99, k101), and the total abundance of NFκB (tot NFκB); and receptor module parameters, which consist of receptor synthesis rates (rcp syn, k54 for TNF, k68 for Pam, k85 for CpG, k35 for LPS, k77 for pIC), degradation rates of receptor-ligand complexes (C deg, k56, k61, k64 for TNF, k75 for Pam, k93 for CpG, k44 for LPS, k83 for pIC), and endosomal import rates (endo, k87 for CpG, k36 and k40 for LPS, k79 for pIC). We then conducted a one-dimensional sensitivity analysis to examine changes in NFκB dynamics due to variations in these parameters. Compared to the default values within the model (black curves in Fig. S1G), each parameter is increased or decreased by $10^{\pm 0.1}$ to $10^{\pm 1}$ times (10 values distributed logarithmically along increasing or decreasing direction, shown as blue and red curves in Fig. S1G). The ODE model was solved in MATLAB using the function 'ode15s' for each virtual cell across two phases: initially driving the virtual cell to a steady state, followed by a second phase of adding specified stimulation. All results were visualized using MATLAB.

## Fitting of parameter distributions to single-cell data

Non-linear mixed-effects model (NLME) was applied to estimate the variations in parameters or initial species values responsible for generating heterogeneous trajectories. The non-linear function in the NLME was modeled by the 52-dimentional ordinary differential

equation of the NFκB signaling network (Supplementary Dataset 2), using the nuclear NFκB activity as the model output $f(t, \psi)$, which is corresponding to the experimental observation $Y$:

$$f(t_i, \psi^{(m)}) = (I\kappa B\alpha N F\kappa Bn(t_i, \psi^{(m)}) + N F\kappa Bn(t_i, \psi^{(m)})) - shift \quad (1)$$

for cell $m = 1, \ldots, M$ and time points $i = 1, \ldots, T$. To quantify the difference between experimental data and simulation data, the experimentally observed NFκB activities (fluorescence intensity) was rescaled to SI Units. Based on prior models[29], the nuclear NFκB concentration was assumed to range from 0.04 μM to 0.30 μM, and we scaled the 90th percentile response to 100 ng/mL LPS to 0.30 μM nuclear NFκB. This resulted in an NFκB A.U. to S.I. scaling factor of 0.0313 (see Supplementary Notes for details). To avoid overfitting, within the NFκB network, 6–7 parameter for each ligand pathway were fitted (Supplementary Dataset 2), which were subject to both fixed effects and random effects. Logit-Normal distribution is applied to constrain the parameter distribution to biologically feasible ranges $(\psi_{lower, l}, \psi_{upper, l})$, $l = 1, 2, 3, \ldots, 6(7)$. For an observation at time $t_i$, we have

$$y_{mi} = f(t_i, \psi^{(m)}) + (a + f(t_i, \psi^{(m)})a)\varepsilon_{mi} \quad (2)$$

$\varepsilon_{mi}$ is the error term. All individual parameters $\psi^{(m)}$ follow the logit-Normal distribution:

$$\varphi^{(m)} = \log\left(\frac{\psi^{(m)} - \psi_{lower}}{\psi_{upper} - \psi^{(m)}}\right) \text{i.i.d.} \sim \mathcal{N}(\mu, \Omega) \quad (3)$$

Where $\varphi^{(m)}$ is the logit transform of $\psi^{(m)}$. The population parameters for estimation are $\theta = \{\mu, \Omega, a, b\}$, and note that individual parameters $\psi^{(m)}$ (or $\varphi^{(m)}$) are latent variables. This model was solved by maximum likelihood estimation (MLE). The log-likelihood function is given by (see Supplementary Notes for details):

$$
\begin{aligned}
\log(p(Y, \psi | \theta)) = & -\sum_{m=1}^{M}\sum_{i=1}^{T} \log(a + f(t_i, \psi^{(m)})b) \\
& -\frac{1}{2}\sum_{m=1}^{M}\sum_{i=1}^{T}\left(\frac{y_{mi} - f(t_i, \psi^{(m)})}{a + f(t_i, \psi^{(m)})b}\right)^2 \\
& -\frac{k+T}{2}M\log(2\pi) - \frac{M}{2}\log(|\Omega|) \\
& -\sum_{m=1}^{M}\frac{1}{2}(\varphi^{(m)} - \mu)^{\mathsf{T}}\Omega^{(-1)}(\varphi^{(m)} - \mu) \\
& +\sum_{l=1}^{k}\log(\psi_{upper, l} - \psi_{lower, l}) \\
& -\sum_{m=1}^{M}\sum_{l=1}^{k}\log(\psi_{upper, l} - \psi_l^{(m)}) \\
& -\sum_{m=1}^{M}\sum_{l=1}^{k}\log(\psi_l^{(m)} - \psi_{lower, l})
\end{aligned}
\quad (4)
$$

After we estimated population-model parameters $\hat{\theta} = \{\hat{\mu}, \hat{\Omega}, \hat{a}, \hat{b}\}$, we then estimated the individual parameters $\hat{\psi}^{(m)}$ via the Maximum a posteriori estimation (MAP):

$$\hat{\psi}^{(m)} = \text{argmax}_{\psi^{(m)}} p(\psi^{(m)} | Y_m; \hat{\theta}) \quad (5)$$

More about the calculation details are in Supplementary Notes. To solve above MLE and MAP, Software Monolix (https://monolix.lixoft.com/) was applied to fit the model to the experimental data. As the TNF condition is the only one that shows strong oscillations and Pam non-oscillatory dynamics are representative of the four PAMP ligands, the

common core module was first parameterized with these two conditions. Then the selected parameters (see method Parameter selection and sensitivity analysis) in each receptor module were estimated using all doses (3 doses for TNF, Pam, and pIC, 5 doses for LPS and CpG (Table 1). The sample size for the parameter estimation of five ligands are: 1809 cells for TNF, 1715 cells for Pam, 1495 cells for LPS, 1558 cells for CpG, and 1763 cells for pIC. All other parameters are constants and from literature[21]. Then each individual parameters were applied to simulate the single-cell NFκB signaling activities using the 52-dimensional ODE (see Supplementary Notes for details).

## Model performance evaluation

We employed three methods to evaluate the performance of our model simulations. (1) For each measured cell, we calculated the root mean square deviation (RMSD) between the model-simulated and experimentally measured NFκB trajectories. (2) We calculated six signaling codons of NFκB dynamics −Speed, Peak, Duration, Total, EvL, and Osc −that encode the stimulus information (Adelaja et al., 2021), to serve as quality control metrics (see Supplementary Notes for details). The distributions of these codons were computed from both simulated and experimental datasets to quantify the dynamic features of NFκB activities across different stimulations. The L2 Wasserstein distance was used to measure the similarity between the codon distributions of the model simulations and experimental measurements under the same conditions. Additionally, the Wasserstein distances among signaling codon distributions from different conditions in the experiments or simulations were calculated to assess the stimulus-response specificity. (3) We applied a random forest classifier using signaling codons to predict the ligand identity for both experimental data and simulated data derived from fitted parameters or sampled parameters (See Method-Generating new dataset of NFκB trajectories). The classifier's parameters were optimized through grid searching, and its performance was assessed using confusion matrices and five-fold cross-validation. The RMSD calculations and signaling codon analysis were conducted in MATLAB, while the random forest classification was implemented in Python using the class 'RandomForestClassifier' within the 'sklearn.ensemble' module of the 'scikit-learn' package.

## Generating new dataset of NFκB trajectories

To generate heterogeneous single-cell NFκB trajectories in response to extended doses of ligand stimulation, we first sampled single-cell signaling network parameters, then simulated NFκB trajectories using the sampled single-cell parameters. Firstly, two parameter sampling approaches were tested. The first was sampling parameters from the estimated population-level statistical model:

$$\text{logit}(\psi) \text{ i.i.d.} \sim \mathcal{N}(\mu, \Omega) \tag{6}$$

We then tested bootstrapping approach: sampling from the non-parameterized distribution which are composed of the inferred single-cell parameters derived from the experiments:

$$f(\psi) = \sum_{m=1}^{M} \frac{1}{M} \delta\left(\psi - \hat{\psi}^{(m)}\right) \tag{7}$$

where $M$ represents the total number of cells across all dosage conditions for that specific ligand. Secondly, the sampled heterogeneous single-cell parameters were applied to simulate the heterogeneous single-cell NFκB signaling trajectories using the 52-dimensional ODE (see Supplementary Notes for details). Simulation results showed that the bootstrapping approach replicates the experimental NFκB dynamic feature distributions, while sampling from the parameterized distribution showed significant discrepancies with the experimental results (see Supplementary Notes for details). Thus, the bootstrapping approach was utilized to generate new datasets of NFκB trajectories.

## Maximum mutual information calculation

To quantify the signaling pathway channel capacity, the mutual information between the stimulus condition $S$ (different doses or ligands) and the NFκB signaling pathway were calculated. For stimulated single cell under condition $S$, the experimental or simulated NFκB trajectories were decomposed into six signaling codons $R$ (informative dynamic features). The mutual information between $S$ and $R$ is

$$I(R; S) = H_{diff}(R) - H_{diff}(R|S) \tag{8}$$

where $H_{diff}(X) = -\int_{-\infty}^{+\infty} f(x)\log_2(f(x))dx$ is defined as the differential entropy. The mutual information were estimated using binless strategy[21,25,47], via k-nearest neighbor estimator for the continuous variable probability density estimation when calculating the Shannon different entropy[62]. Then mutual information was maximized over all possible stimulus distributions $p(S)$ to quantify the Channel Capacity:

$$C(R; S) = \max_{p(S)} I(R; S) \tag{9}$$

This was implemented by ANN MATLAB Wrapper (ver 1.2) in MATLAB.

## Modeling the IκBα^S/S genotype

To model this IκBα promoter mutant, the NFκB-regulated IκBα transcription rate (parameter k), was reduced by a four-fold[23]. To generate the heterogeneous single-cell trajectories, the same approach as for a new dataset for WT was applied. The first step was to sample single-cell parameters, whose distribution was inferred from the WT dataset. Then the sampled heterogeneous single-cell parameters, with the NFκB-regulated IκBα transcription rate reduced by four-fold change, were applied to simulate the heterogeneous single-cell NFκB signaling trajectories using the 52-dimensional ODE distribution.

## Denoise modules to quantify information gain within NFκB signaling network

For heterogeneous single cells, 6–7 parameters of the NFκB model (including receptor, adaptor, and NFκB-IKK core modules) were distributed from the distributions inferred from the experimental dataset and then employed to simulate single-cell trajectories. To denoise a specific module within the NFκB signaling network, the parameters within the specific module were set to one set of parameters (the representative cell or a random virtual cell parameter values), i.e., collapsing the parameter distribution to single set of parameter values for the denoised module, while the parameter distributions in other modules remained the same as inferred. Such denoised-module single-cell parameters are then applied to simulate heterogeneous single-cell trajectories. Model simulations were implemented through the Matlab 2020a with the ode15s solver.

The denoised-module NFκB trajectories were decomposed into six signaling codons, and then applied for calculation of mutual information between stimuli (S, here representing different ligands) and the response (R, informative dynamic features of NFκB trajectories) using the same formula as previous section:

$$I(R; S) = H_{diff}(R) - H_{diff}(R|S) \tag{10}$$

The mutual information was maximized over all possible stimulus distributions $p(S)$ to quantify the Channel Capacity after denoise specific modules within the signaling network. Then different denoising-module results were compared with the channel capacity of the original signaling network (without denoise).

## Simulating heterogeneous single-cell responses to different stimuli

Based on inferred single-cell parameters (Methods – Single-cell model fitting), we developed an approach to generate heterogeneous virtual single-cell NFκB network parameters via bootstrapping sampling and core-module parameter matching, as described in further detail in the subsequent paragraphs. The generated heterogeneous single-cell NFκB network parameters were applied to simulate the heterogeneous single-cell NFκB signaling trajectories in response to different ligands using the ODE.

Since the single-cell parameters inferred from the experimental dataset include only one receptor module and the core module, the parameters for the other four receptor modules were missing and must be inferred to reconstruct the complete NF-κB signaling network for each single cell (Fig. S5A). To generate 1000 heterogeneous virtual single-cell NFκB networks, our approach began by sampling 200 sets of parameters from those inferred from experimental dataset for the first ligand and core module. This allowed us to identify the first receptor module parameters along with the core module for each virtual single cell. To determine the parameters in the other four receptor modules for the same individual cell, we assigned each receptor module parameters from the single cells that sharing the same core module parameters (or most similar). This process generated 200 heterogeneous single-cell NFκB signaling networks. We then repeated this workflow for each receptor module, using it as the initial sampling module in turn, until the process was completed and 1000 virtual single-cell NFκB signaling networks were generated.

For example, we first sampled 200 single-cell parameters inferred from TNF-stimulated cells. For each sample of these 200 cells, we obtained the TNF module parameters along with core module parameters, but still needed the parameters for the LPS, CpG, Pam, and pIC receptors. To assign the LPS receptor parameters to each of these sampled 200 cells, we calculated the differences between the sampled core module parameters and the LPS-stimulated single-cell core module parameters, selecting the LPS-stimulated cell with the smallest difference and assigning its LPS receptor module parameter value to the sampled cell. We repeated this process for the CpG, Pam, and pIC receptors. This approach allowed us to generate 200 virtual single cells with heterogeneity across all five receptor modules, using TNF as the first sampled ligand. Then we repeated this for LPS as first sampled ligand, and so on.

## Statistics and reproducibility

Experimental data used for model fitting were representative of ten replicates, some of which were presented in previous publications[5,21,23,24]. The TNF-stimulated samples in this study—without the soluble TNF receptor II—exhibited NFκB signaling dynamics consistent with previously published independent datasets, including Adelaja et al. 2021 (one biological replicate)[21], Singh et al. 2024 (three biological replicates)[24], Rahman et al. 2024 (three biological replicates)[23], and Luecke et al. 2024 (two biological replicates)[5]. This agreement among multiple datasets indicates the reproducibility of the experimental data used for the mathematical model fitting presented here. Each experimental condition yielded data for 176–744 cells as indicated in Table 1. The total number of cells (sample size) processed was constrained by the cell density in the culture and the number of regions that could be captured with a 5-min imaging interval. No statistical method was used to predetermine sample size but the sample size was the result of the image analysis using MACKtrack (https://github.com/Adewunmi91/MACKtrack). For the fitting and simulation dataset under TNF conditions, any model simulations exhibiting a low coefficient of variation (lowest 67% of CV) were excluded from further analysis, along with their corresponding experimental data, as they did not reliably capture the noisy sustained oscillations observed experimentally.

## Reporting summary

Further information on research design is available in the Nature Portfolio Reporting Summary linked to this article.

## Data availability

The experimental and simulation data generated in this study have been deposited in the Dryad database under https://doi.org/10.5061/dryad.8cz8w9h3d. Source data are provided with this paper.

## Code availability

The code used to develop the model, perform the analyses and generate results in this study is publicly available at GitHub at https://github.com/Xiaolu-Guo/Virtual_single_cell_NFkB, under MIT license. The specific version of the code associated with this publication is archived in Zenodo and is accessible via https://doi.org/10.5281/zenodo.15062580. The original codes for image analysis, for representative cell NFκB simulation and signaling codon calculation is deposited to GitHub https://github.com/Adewunmi91/nfkb_model and https://github.com/Adewunmi91/MACKtrack[21].

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

## Acknowledgements

The work is supported by National Institutes of Health (RO1AI173214) to A.H. A.S. acknowledges support from the National Institutes of Health (NIH), National Institute of General Medical Sciences training grants T32GM008185 and T32GM008042. We thank Eric Deeds for discussions on information theory, and critical reading of the manuscript. We thank Stefanie Luecke for measuring macrophage volumes and Xiaofei Lin for advice on parameter selection and signaling codons calculation. We thank Helen Huang, Xiangting Li, Allison Schiffman, and Maxim Shokhirev for critical reading of the manuscript.

## Author contributions

A.H. and X.G. designed the research. X.G. developed the computational workflow and generated and analyzed all simulation data. A.S. and R.W. provided technical advice. A.A. generated and X.G. analyzed experimental data. X.G. and A.H. wrote and all authors edited the manuscript.

## Competing interests

The authors declare no competing interests.
