## [Transparent Peer Review file · Nature Communications]

Modeling heterogeneous signaling dynamics of macrophages reveals principles of information transmission in stimulus responses

Corresponding Author: Professor Alexander Hoffmann

Version 0:

Reviewer comments:

Reviewer #1

(Remarks to the Author)

In this paper, the authors use a detailed mechanistic model to explore the ability of single macrophages to discriminate different types and doses of immune ligands, such as cytokines or bacterial/viral PAMPs. To do so, they modified existing mathematical models of NF κ B signaling by fitting to single cell trajectories of RelA nuclear translocation in mouse macrophages stimulated with different doses and types of cytokines (TNF) and PAMPs.

This work builds meaningfully on landmark previous work (Adelaja et al., *Immunity*) by extending their understanding of ligand discrimination via NF κ B signaling codons from population averages to single cells. In doing so, they estimate that groups of 2-4 macrophages are sufficient to achieve perfect ligand discrimination. A less novel conclusion of the paper is that receptors are the key bottleneck where information loss occurs during signaling. Although the major conclusion (that small macrophage groups can compensate for one another's errors) is intriguing and biologically interesting, this reviewer is not convinced that the data fully supports this claim. Further, several discrepancies between this paper and Adelaja et al. cast doubt on the validity of their math model and their conclusions. Overall, this paper makes good use of a mechanistic model to draw interesting conclusions, yet those conclusions are – in some cases – not novel or in conflict with their previous work.

Points:

- Lack of novelty in a major conclusion: The authors conclude that receptors are a key site of information loss in signaling, which is not novel and has been shown by multiple other groups (PMID: 34301608, 23908238, 21921160), only one of which is cited. The utility of their particular model in drawing this conclusion again is unclear and, in my opinion, bloats the manuscript, diluting the more interesting message. The authors should clarify what is unique about this conclusion – as opposed to what has already been shown. If it is not unique, I recommend reducing the size of figures 2 and S2 and the accompanying text, and employing this figure as a validation of the model, rather than novel finding.

- Discrepancy between their single cell model and experimental data from Adelaja et al., cast doubt on the model validity: The authors conclude that the I κ B-NF κ B module is extremely robust to parameter variation in terms of preserving SRS. This is confusing given the results of their *Immunity* paper. There, mutations in the I κ B-NF κ B core module led to significant confusion of ligands that (presumably) drove autoimmune pathogenesis in the Sjogren's model (Fig 4). Their previous model implicated components of the core module – such as IKK – as key for oscillatory behavior and hence SRS. The extent of ligand confusion predicted from their single cell model is quite modest in comparison (Fig 5E-H). The authors must clarify these discrepancies between their previous experimental data and model, and the model described in this manuscript.

- Conceptual discrepancy between a major conclusion and their own prior work: In this paper, the authors conclude that small consortia (2-4) of macrophages are sufficient to achieve nearly perfect ligand discrimination. This is expected to increase robustness and mistakes by single cells. Given that their simulated I κ BaS/S virtual cells discriminate nearly as well as the wild type virtual cells (Figs 5E-H), with errors distributed somewhat randomly, shouldn't macrophage consortia be sufficient to compensate for the slightly reduced discriminatory power of I κ BaS/S virtual cells? This would negate their earlier conclusion that ligand confusion underlies autoimmune pathology because cell collectives in vivo should confer robustness to the errors made by individual Sjogren's macrophages. Rather than generating a dataset of low diversity (Fig

6), the authors should instead compare the error correlations and consortia size necessities between lkbBaS/S and wild type virtual cells. If they are not sufficiently different, it casts doubt on their conclusion that macrophage consortia overcome errors given their prior experimental results/modeling.

- Lack of experimental evidence for a major conclusion: Their model generates a prediction that individual cells can compensate for errors made by their brethren because the types of errors made are random. While I appreciate that a direct testing of this hypothesis is outside the scope of this study, it would be helpful for the authors to suggest – at least – an experiment that could be employed to test their hypothesis. If testing it is impossible, the conclusion will remain speculation.

(Remarks on code availability)

The code is very extensive, but appears well-organized and annotated enough to make the model usable to others in the field. It is transparently presented.

Reviewer #2

(Remarks to the Author)

The focus of this manuscript is in the broad area of NFkB signaling, a transcription factor known to be critical in immune cell response to stimuli, often pathogenic or cytokine driven. The manuscript advances our understanding of the single-cell stimulus-specific response through theoretical framework, beyond what is readily accessible through experimental data. The authors expand upon a previously published model from their group to generate a virtual cell and assess the NFkB dynamics in response to various stimuli. Of their main observations, they note that information loss is mainly due to extrinsic noise in the receptor module. Further, they conclude from their simulations that while there may be blind-spots, the heterogeneity in cell populations of as small as 2-6 macrophage consortia is sufficient to achieve perfect stimulus distinction. The manuscript is well written, with demonstration of good scientific rigor. However, there seems to be a lack in novelty or major conclusions in the manuscript which distinguishes the current draft from their previously published work. There are some concerns which could be addressed to enhance its clarity, impact and scholarly rigor.

Major concerns:

1. This paper is mainly reliant on presenting a theoretical framework as a tool for generating synthetic data where experimental data is not easily attainable. However, given the model has already been published by the authors in 2021 (Adelaja et al., 2021), there is no new development in that front which distinguishes this manuscript from their previous work. Even the choice of parameters has been derived from previous studies of the authors and others.
2. The authors suggest the current model to be more complex (133 parameters, 52 species) than some of the other works in the area (52 dimensions vs ~10 dimensions). Yet only a subset of biochemical constants is chosen for parameterization. The authors should provide more rationale behind the choice of these parameters. Is the model biased by the selection of these parameters, and how accurately does it reflect experimental data if different set or additional parameters were chosen? It appears the rest of the biochemical constants come from prior literature. Are these interchangeable for the two cell descriptions of BMDM and hMPDMs, given their arguments in an earlier study about the heterogeneity of BMDM population as opposed to the hMPDMs?
3. It is also unclear why the authors chose two different cells in this study and how some of their conclusions about the macrophage consortia would change if they utilized only one cell line. Furthermore, it is quite confusing where they utilized the hMPDMs experimentally, as stated in the methods section. From the main text, it appears that simultaneous ligands were introduced only for the simulation case and not experimentally (Fig5). The authors should revisit this and make it clear.
4. Expanding on the previous points, is the model universally applicable or do all the parameters need to be re-computed for another cell line/ model system? If the latter, then the process becomes tedious and limits the utility of the framework. Furthermore, if sequencing experiments can provide similar inferences about NFkB dynamics as shown by the authors in their previous work (Sheu et al., 2023), isn't that a more direct and informative approach?
5. From Fig1C, it can be noted that the number of cells vary between 176 to 744 which is more than 4-fold different. Can the authors comment on how this impacts statistically on the model parametrization? Do the authors speculate a similar sample size for all the conditions may alter the model outcomes?
6. Some of the choices seem subjective as they are reliant on visual inspection of trajectories. For eg. RMSDs threshold of 0.03 for acceptable fits
7. The authors compared the model simulation against the live cell macrophages using machine learning classification. What is the rationale behind only outputting the ligand information at high doses alone, and not test the other conditions?
8. TNF condition seems to be most distinguishable in the ML classification in Fig 2D. It is understandable that TNF produces oscillatory profile which is distinguishable from other ligands. However, it is also to be noted that the core module parameters were first fitted to TNF stimulated dataset. Could this have led to a bias in this regard?
9. The results of Fig3A, while tedious, can be verified experimentally. The authors should demonstrate at least a few examples outside of the training set, where their theoretical predictions match with experimental observations.
10. The explanation of the discrepancy of TNF MI between theoretical vs experimental observations remains unclear. Furthermore, the authors describe in discussions that the information transmission via duration and EvL signaling codons could be due to intrinsic noise within the lkb α loop. This does not reflect in the current information loss schematic of the model. The authors need to address this section more clearly.
11. Can the authors suggest how would they hypothetically validate testing their conclusion about the loss of information in the receptor module experimentally?
12. As brought up in an earlier point, the section describing Fig5 seems unclear. For eg., which cells were used for the experiments vs simulations, were the initial conditions same for the parameterized virtual cell, how is the heterogeneity introduced etc.? Please address this section more to avoid confusion for the readers.
13. The generation of new data sets for this section relating to Fig5 again relied on high doses of the ligands. Why are other

doses not considered in the analysis?

Minor concerns;

1. Check the references again as there are some inconsistencies in the citation formatting.
2. The authors setup the introduction describing how there are outstanding questions that can't be addressed experimentally. However, it seems that only correlation of population level SRS with specificities of individual cells is unanswerable with experiments alone while the true channel capacity is already computed from experiments. In this regard, finer dose gradations can also be experimentally interrogated.
3. Maybe mention the total number of biochemical constants used for parameterization in the main text (at least the range) for clarity rather than just reference to supplementary figures for this detail.

(Remarks on code availability)

Reviewer #3

(Remarks to the Author)

The manuscript by Guo et al. addresses the important question how immune cells can reliably process information about stimuli despite the heterogeneity in cellular signaling. This is done by combining live-cell imaging of the transcription factor NFkB in mouse macrophages, mathematical modeling using the established model of Adelaja et al 2021 and a comprehensive data analysis pipeline. The approach is used to demonstrate how well the cellular NFkB response can be distinguished between stimuli, how the stimulus response specificity (SRS) is limited by noise and where information loss due to noise occurs. The parametrized single cell models are finally used to evaluate the SRS of 1000 simulated cells that were not investigated experimentally. This overall highly interesting approach can serve as a very good example how mechanistic modelling combined with data analysis can predict cellular responses to stimuli, doses and perturbations not measured. However, there is a number of questions with respect to the limits of the approach and how far the results expand the insights of the preceding Adelaja et al. paper.

Major points:

- The mathematical modeling uses the model of Adelaja et al. 2021 and reparametrizes it based on extensive single cell data. The earlier paper already included single cell data measurement based on an identical workflow to identify stimulus-response specificity (SRS), therefore it is not really clear what set the new approach apart from Adelaja et al. and what additional insights are gained. The workflow scheme 1B is identical to Fig 1B in the earlier paper.
- Some formulations could be misinterpreted as a development of a new model (line 80 'development of mathematical model simulations', abstract 'we developed a mathematical model'), this should be clarified to 'model simulations' or 'simulations of the mathematical model'.
- Given the highlighted importance of the negative feedback regulation it is not clear why the regulation by A20 that forms a second negative feedback loop within the IKK+Nfkb core system is not included into model. How would that feedback effect the NFkB dynamics and signaling codons? This needs at least to be discussed based on the established role of that feedback on the dynamics of core models with A20 feedback (line 49-52).
- The introduction is not balanced according to the contribution of other groups in the field, these need to be acknowledged more carefully. Highly dynamic NFkB behavior, in particular oscillations have first been shown by M Whites lab (Nelson et al Science 2004, Ashall et al. Science, 2009), also by MH Sun et al. Plos One 2009 and others (lines 18-22). Some of these references are cited later in the context of 'cell-to-cell heterogeneity casting doubt' but those references are missing in the reference list (while several of the group references occur even twice). Also the discussion of informative signaling features was started early by S Gaudets group (RE Lee et al. 2014 MolCell) and should be cited (line 26-28).
- The generated experimental data set is described as 'using five ligands at 3 doses with up to 800 single cell NFkB trajectories for each condition' (line 111-114). The details in Figure 1C show that the cell numbers vary between 176 and 744 and should be described like that. Moreover, the possible impact of different cell numbers in the different conditions, especially the lower numbers for LPS and CpG stimulation, should be discussed. Additionally, the goodness-of-fit is described as 'with a median of ~0.02 for each of the 15 stimulus conditions' (line 150) and 'a threshold of 0.03 provided for acceptable fits (Figure S2B), which were satisfied by ~80% of the trajectories' (line 152-153). However, Figure S2A and S2B show, that the median ranges from 0.01 to 0.03 and the fraction of trajectories with acceptable fits range from 60% to ~95% for the different conditions. This should be described more precisely since the data shows a different goodness-of-fit for the individual conditions. Particularly, it seems that the TNF low condition can be much better reproduced than e.g. the LPS medium condition.
- For the data fitting a sequential approach is used. What is the motivation for that? Does the sequence of selected data impact the results, e.g. have TNF and PAM data a higher weight in that approach because of their earlier selection? What would be the difference to applying a multi-experiment fitting approach, e.g. Fröhlich et al. (<https://doi.org/10.1038/s41540-018-0079-7>)? Moreover, the statement about the simulated trajectories that were used for the analyses ('For TNF conditions, cells with simulated trajectories in the top 33% of the coefficient of variation (CV) were selected for further analysis, including evaluation of the model simulation and its applications.', last sentence in the supplement, section 5.3) is unclear and need to be described in more detail and explained why only TNF conditions are included in the selection process.
- While the statement that 80% of single cells distinguish 9 to 10 stimulus pairs is highly intriguing, it strongly depends on the chosen threshold. In Figure 5C the distribution of L2 distances between the given signaling codons is shown and two examples of corresponding trajectories. However, it is unclear how strong the difference between individual signaling codons is and more importantly how this translates to the transcriptional response.
- Finally, the authors claim that 'neighboring cells compensate for each other's blind-spots' (line 351-352). This statement seems too strong given that in a multicellular context, one expects that the cellular response in this environment is a

combination of the responding cells, especially in the context of cytokine excretion. Cells that cannot distinguish two signals, could therefore dilute the response of cell distinguishing the signals. This should be clarified.

- How does the block of para- and autocrine TNF signaling impact the results on small consortia of macrophages were the authors conclude that diversity among the single cells of a consortia enables perfect SRS? A discussion how far the data and analysis represent a case of modified interactions and diversity is required.

There is information missing:

- table S1 und S2
- Fig 5I is missing

Minor points

- References to figures are incorrect in line 152 and 153: reference to Figure S2B should be Figure S2A, right panel, and the reference to Figure S2C should be Figure S2B.
- The statement in line 328 'for all stimulus pairs the 50th and 90th percentile of L2 were diminished' is incorrect. For CpG vs Pam it is increased.
- The statement 'The I κ B α S/S genotype reduced the proportion of cells being able to distinguish all stimulus pairs by 5% compared to WT' (line 330) is unclear. Is the 5% reduction related to the average of all conditions? The following given percentages in line 331-332 are also not clear where they come from. Because comparing the percentages of individual conditions for the I κ B α S/S genotype and WT show that for the I κ B α S/S genotype confusion can also be reduced e.g. for pIC vs PAM (16% to 13.1%).
- Figure number missing in supplement in section 6.1

(Remarks on code availability)

The code of the modeling and analysis is available and the documentation in the supplement is comprehensive.

Reviewer #4

(Remarks to the Author)

(Remarks on code availability)

Version 1:

Reviewer comments:

Reviewer #1

(Remarks to the Author)

The authors have reasonably addressed all of my previous claims. I still argue that the progress past their past paper is somewhat incremental, but I think they were responsive to everything raised by myself.

(Remarks on code availability)

Reviewer #2

(Remarks to the Author)

The authors have amended fig1 and corresponding text in the manuscript to highlight the differences. However, it is evident that the underlying model is the same framework used in Adelaja et al. (2021). In the previous work, the fit is done on a representative profile (the choice of the representative case is not clearly described) while this work considers the different profiles which represents the heterogeneity. In the rebuttal, the authors mention it was the norm to only model a representative profile but other studies model either using an average expression profile (by the authors themselves in DOI: 10.1126/science.1071914) or some that have also considered single-cell data (DOI: 10.1126/science.1164860). Moreover, several studies have demonstrated single cell models with heterogeneity through different parameter selection (doi.org/10.1038/nature09145, doi.org/10.1038/ncomms12057, doi.org/10.1073/pnas.0913798107, doi.org/10.1186/1752-0509-7-45), hence the claim of it being done for the first time in this manuscript, is misleading. Further, the authors acknowledge that first fitting the core module to TNF and PAM may favor their fits. In other words, does it accurately capture single cell heterogeneity in cellular response. It is mentioned that fits are not worse for other ligands by this approach but is not shown.

(Remarks on code availability)

Reviewer #3

(Remarks to the Author)

We thank the authors for their answers, modifications and additional analyses, which explain and clarify many aspects of the manuscript. Most of the major points are now resolved. In particular, the modified Fig 1a and the revised text parts clarify what is new compared to Adelaja et al. (2021) (point 1).

Point 7. The presentation of the investigation for a different threshold shows the robustness of the results. The reviewer agrees that the additional data provide a wealth of information and it would be highly interesting how this translates into gene expression.

Point 8. The clarification of the underlying assumptions makes the line of argumentation now more clear.

There are three open points

Point 3. The authors argue that the A20 transcriptional feedback is not important for the early NFkB response. While this has been demonstrated for short time frames (Cruz et al., 2021), it might not be correct at later time points, as Werner et al., 2008 showed that a A20 KO can change the single cell dynamics of decaying NFkB after around 60 minutes for TNFa constantly present. Since in the manuscript single cells are partly tracked until 8h, A20 might have an impact depending on how long TNF is present in the experiment and the model. It would be good to clarify this point.

Point 4. A reference list is missing in the revised paper version.

Point 10. There are still no tables S1 and S2.

(Remarks on code availability)

Reviewer #4

(Remarks to the Author)

(Remarks on code availability)

Editor comments:

You will see that, while the reviewers find your work of interest, they raise substantive concerns that cast doubt on the advance your findings represent over earlier work and the strength of the novel conclusions that can be drawn at this stage.

We thank you for functioning as the editor for this manuscript, and are pleased that the reviewers found the work of interest such that they provided detailed reviews.

We believe we can compellingly address concerns about the novelty of the work and the strength of the evidence for the novel findings and conclusions.

1. Novelty: (1) In this paper we present for the first time a mathematical model simulation that captures the heterogeneity of single-cell NF κ B signaling dynamics within a population of macrophages. Reviewers rightly mention that our Immunity paper in 2021 also presented a mathematical model of NF κ B signaling dynamics, but it is important to recognize that this was capable of only a single signaling trajectory for each simulation and did not capture the heterogeneity. With the advance reported here we could actually match simulations to specific measured cells. We showed that the simulations fit the measured data very well, but we also showed that the simulations of all cells recapitulate the stimulus-response specificity (SRS) of the experimental data in terms of the pattern of Wasserstein distances and the machine learning classification analysis. It may be worth pointing out that this model simulation is not only the first time that we can now study the role of heterogeneity in the fidelity of NF κ B signaling, but it is to our knowledge the first time that such studies are presented for any mammalian signaling pathway.

(2) As a first step, with this new model simulation we could now address where information loss occurs within the pathway. To this end we developed a new workflow involving the denoising of different modules or molecular mechanisms. What is gratifying is that the results from our simulations match the conclusions drawn by a previous publication that used experimental measurements; our analysis goes further that what was experimentally possible but the prior work provides confidence that our conclusions are sound.

(3) The primary novel contribution of the manuscript is the derivation of the stimulus-response specificity of single cells (scSRS). Ultimately, macrophages function as single cells and stimulus-response specificity is a key hallmark of healthy macrophages. However, SRS has only ever been measured at the population level. That is because a cell is no longer naïve after being stimulated once, and hence cannot be stimulated twice or more to determine its scSRS. With the mechanistic model in which the degree and source of heterogeneity is correctly recapitulated, we are able to infer scSRS for the population of cells. This revealed that cells actually differ remarkably in which stimulus pairs they are unable to distinguish. The result makes sense given that the key information bottle neck is the receptor, which was experimentally supported. So in this way the finding is actually well supported, but we are also able to suggest some experimental strategies to examine this point in the future as reviewer 1 is suggesting.

We recognize that the presentation of Figure 1 in particular was suboptimal, undercutting the actual novelty and impact of the work. We have made appropriate revisions to avoid the confusion and clarify the novelty:

- a. *we have replaced Figure 1A-B with a new workflow figure that clarifies the novelty and how the present work relates to published work (address reviewer 2, comment 1; reviewer 3, comment 1).*
- b. *we have improved the introduction emphasizing the importance of cellular heterogeneity and its role in immune response to pathological threats and the*

limitations of experimental studies and prior modeling studies in addressing key question. For example, no one has been able to measure single-cell responses to different stimuli, making it impossible to determine the stimulus-response specificity of individual cells (address reviewer 2, comment 1; reviewer 3, comment 1).

- c. *we have extended the results figures 3, 4 and 6 and added more explanation to clarify what is novel and what is aligned with the literature (address reviewer 1, comments 1-3, reviewer 2, comments 9, 11).*

2. Strength of novel conclusions:

This is a critical point and we have taken care to better summarize the evidence and quantitative evaluation of the key conclusions.

- a. In model construction and parameterization we have explained the methods more clearly (address reviewer 1, comment 1, reviewer 2, comment 12, reviewer 3, point 6).
- b. We have tested different settings of parameters and test the robustness of the model outcomes for different sample size (address reviewer 2, comments 2-8, 13, reviewer 3, comment 6).
- c. For figure 3, we have compared model prediction on finer doses response (beyond training range) with published experimental dataset (address reviewer 2, comment 9).
- d. We have compared simulation results of I κ B α promoter mutant to published experiments of I κ B α promoter mutant, showed that model prediction is qualitatively consistent with experiments (address reviewer 1, comments 2-3).
- e. We have discussed potential experiments to measure scSRS to evaluate the predictions from our model simulations (address reviewer 1, comment 4).
- f. We have considered alternate scenarios by which consortia of cells function. We have also identified differences in the performance of consortia of WT and I κ B α mutant macrophages (address reviewer 3, comment 9, reviewer 1, comment 3)

3. We have taken care to address all other reviewer points in other revisions:

- a. we have improved the corresponding parts of intro & results & discussion and include additional citations as suggested by reviewers (reviewer 3, comments 2-5).
- b. we have discussed how para- and autocrine mechanisms might play a role in macrophage consortia of SRS (address reviewer 3, comment 9)
- c. we have revised all minor points of fixing reference format, clarify the missing/misleading information (address reviewer 2&3, minor comments)

Reviewers' comments:

Reviewer #1 (Remarks to the Author):

This work builds meaningfully on landmark previous work (Adelaja et al., Immunity) by extending their understanding of ligand discrimination via NF κ B signaling codons from population averages to single cells. In doing so, they estimate that groups of 2-4 macrophages are sufficient to achieve perfect ligand discrimination. A less novel conclusion of the paper is that receptors are the key bottleneck where information loss occurs during signaling. Although the major conclusion (that small macrophage groups can compensate for one another's errors) is intriguing and biologically interesting, this reviewer is not convinced that the data fully supports this claim. Further, several discrepancies between this paper and Adelaja et al. cast doubt on the validity of their math model and their conclusions. Overall, this paper makes good use of a mechanistic model to draw interesting conclusions, yet those conclusions are – in some cases – not novel or in conflict with their previous work.

We thank the reviewer for their careful assessment of the manuscript and for recognizing our math model-based approach enables analyses “from population averages to single cells” and estimates the group size needed “to achieve perfect ligand discrimination”. In our responses below, we further clarify how our model aligns with Adelaja et al. (2021), and revised the corresponding manuscript to address these. Thus, this supports and validates our models’ capacity to recapitulate experimental data and make reliable predictions.

Points:

- Lack of novelty in a major conclusion: The authors conclude that receptors are a key site of information loss in signaling, (a) which is not novel and has been shown by multiple other groups (PMID: 34301608, 23908238, 21921160), only one of which is cited. (b) The utility of their particular model in drawing this conclusion again is unclear and, in my opinion, bloats the manuscript, diluting the more interesting message. (c) The authors should clarify what is unique about this conclusion – as opposed to what has already been shown. If it is not unique, I recommend reducing the size of figures 2 and S2 and the accompanying text, and employing this figure as a validation of the model, rather than novel finding.

Reply: We thank the reviewer for pointing out additional papers that have identified receptors as bottlenecks for information flow. It is reassuring that our conclusion based on our model simulations is aligned with prior studies, and we would like to point out our findings are derived from a novel approach. Our workflow of denoising different modules within the single-cell signaling networks to locate the information loss is a novel approach of assessing information flow. Below are responses point-by-point.

(a) We appreciate the reviewer’s providing more related literatures. In the previous version, we have cited PMID: 34301608, 21921160 (in discussion, Lines 546-552). In the revision, we have also added these two references in the results section as a validation of our approach to locate information loss (lines 361-363). In addition, PMID: 23908238, (Uda et al., 2013) quantified the information flow in growth factor signaling pathways and showing the robustness of signal fidelity under different pharmacological perturbations. We have also added all these three references in discussion to characterize experimental approaches on studying information flow (lines 552-558) (see point c).

(b) In the revision, we have added Figure 4B and corresponding texts in results section (lines 340-342) and method section (lines 765-784, Denoise modules to quantify information gain within NF κ B signaling network) to explain our developed workflow in detail. Based on the well-parameterized single-cell models, we developed a new workflow to quantify the information gain

after denoising different modules. We hypothesize that denoising different modules by collapsing parameter distributions (in that module) to one set of parameter values can reduce the NFkB response uncertainty and thus increase mutual information between NFkB signaling pathway and the stimuli. Using our workflow to systematically “denoise” each module and re-calculate the mutual information validated our hypothesis. Previous literature aligns with our conclusion of locating information loss to receptor module, and thus validates our approach.

(c) We agree with the reviewer that the main conclusion of locating information loss to the receptor module agrees with prior literature. Interestingly, the prior conclusions were based on experimental studies using two analytes (end points of a branched architecture) measured at a single timepoint. Here, we used the opportunities that the mathematical model affords: we could eliminate the molecular noise in key modules and measure the effect on a single analyte. This workflow has not previously been used in the field, because the requisite mathematical model was not available. Importantly, our conclusions are based on considering the full dynamic trajectory. The fact that our conclusions match the prior experimental results is significant in two ways: 1) our model, particularly how the noise is distributed, is further validated, and 2) the prior conclusions are further strengthened given that we include considerations of the dynamic trajectories of stimulus responses. (lines 552-558).

- (a) Discrepancy between their single cell model and experimental data from Adelaja et al., cast doubt on the model validity: The authors conclude that the IκB-NFκB module is extremely robust to parameter variation in terms of preserving SRS. This is confusing given the results of their Immunity paper. (b) There, mutations in the IκB-NFκB core module led to significant confusion of ligands that (presumably) drove autoimmune pathogenesis in the Sjogren’s model (Fig 4). Their previous model implicated components of the core module – such as IKK - as key for oscillatory behavior and hence SRS. (c) The extent of ligand confusion predicted from their single cell model is quite modest in comparison (Fig 5E-H). The authors must clarify these discrepancies between their previous experimental data and model, and the model described in this manuscript.

Reply: We thank the reviewer for their valuable and insightful comments. Prompted by them we have made improvements to the analysis and presentation that clarify that the insights based on Figures 4 and 5 about the the IKK-IκBα-NFκB core module are consistent with, and extend upon, previous literatures. Following are responses point-by-point:

- (a) If the reviewer’s comment “The authors conclude that the IκB-NFκB module is extremely robust to parameter variation in terms of preserving SRS” refers to **Figure 4E (Figure 4D in previous version)**, our response is as follows: **Figure 4E** shows the results of information loss analysis in the WT. When we collapse the parameter distributions of the core module (i.e., IκBα synthesis delay and NFκB total abundance) to a single, randomly selected WT single-cell value, we observe little information loss (**Figure 4E**). Therefore, we conclude that the extrinsic noise within the core module contributes minimally to information loss in the WT. It is important to note that while we collapsed the core module parameters to a single value for all cells, this collapsed value was still from one of the WT cells.
- (b) In the previous version, we introduced genotype IκBα^{S/S} in Figure 4F-G (refer to previous version). We then calculated the mutual information (MI) between the dynamic features of NFκB and the stimuli for both the wild-type (WT) and IκBα^{S/S} genotype. The model predicted an information reduction of 0.1 bits in IκBα^{S/S} genotype compared to WT, which may seem like a modest difference. For reference, the results from Adelaja et al., (2021,

Figure 5E) similarly revealed a difference of approximately 0.1 bits in gene expression channel capacity between WT and mutant.

To further validate our conclusions in comparisons with previous work (Adelaja et al., 2021), in the revision, we applied the similar approach of machine learning classification to investigate the ligand distinction at the population level. Using the experimental data from (Adelaja et al., 2021), we obtained a 29% decrease in accurate LPS predictions and an 5% decrease for PolyIC in the $IkBa^{S/S}$ genotype compared to WT, while our model predicted a 19% decrease in LPS accuracy and a 2% decrease for PolyIC. The experimental data also indicate a 26% increase in false predictions, where TNF is incorrectly predicted as LPS. The model, in turn, predicted a 20% increase in the corresponding false prediction. It is important to note that all these predictions are based on a single change to the NF κ B-mediated $IkBa$ synthesis rate, with all other parameter distributions remaining identical between WT and the $IkBa$ mutant. In actual cells, secondary/indirect effect changes may alter other biochemical mechanisms, but it appears that the model is remarkably reliable in predicting the performance drop of this Sjögren's mutation. All these results are updated in Figure 4H, S4B-C and manuscript (Lines 373-382).

- (c) For Figure 5E-H, our model for the $IkBa^{S/S}$ genotype (validated at population level in Figure 4H and Figure S4B-C) predicted increased single-cell stimulus response confusion. The results show a significant increase in the ratio of cells confusing TNF with LPS—one example of cytokine vs. PAMP confusion—from 8.9% to 15.4%, while confusion among other ligands remained modest (Figure 5F,H). Such confusion increase are robust across different threshold setting ups for confusion (Figure S5G). However, as shown in (Adelaja et al., 2021) Figure 5E, only a modest difference in ligand distinction through gene expression was observed in the $IkBa^{S/S}$ mutant. On the other hand, more dramatic changes could have lethal consequences, such as in the case of $IkBa$ knockout mice, which are non-viable.

- Conceptual discrepancy between a major conclusion and their own prior work: In this paper, the authors conclude that small consortia (2-4) of macrophages are sufficient to achieve nearly perfect ligand discrimination. This is expected to increase robustness and mistakes by single cells. Given that their simulated $IkBa^{S/S}$ virtual cells discriminate nearly as well as the wild type virtual cells (Figs 5E-H), with errors distributed somewhat randomly, shouldn't macrophage consortia be sufficient to compensate for the slightly reduced discriminatory power of $IkBa^{S/S}$ virtual cells? This would negate their earlier conclusion that ligand confusion underlies autoimmune pathology because cell collectives in vivo should confer robustness to the errors made by individual Sjögren's macrophages. Rather than generating a dataset of low diversity (Fig 6), the authors should instead compare the error correlations and consortia size necessities between $IkBa^{S/S}$ and wild type virtual cells. If they are not sufficiently different, it casts doubt on their conclusion that macrophage consortia overcome errors given their prior experimental results/modeling.

Reply: We thank the reviewer for these comments and have added analyses to the manuscript that more clearly show how the new conclusions are consistent with the published work, while representing a substantive extension.

- (1) In Figure 6, we are asking about the diversity of scSRS profiles among cells within the population. That is the reason for generating a low diversity dataset. The analysis shows that such diversity allows cells to compensate for each other's blind spots, thereby achieving perfect distinction as a small consortium. In addition, and separately we are

asking about the role of the I κ B-NF κ B feedback module in in this process, so that we are extending the previous analyses (Figures 4 and 5) with the Sjögren's mutant.

*(2) We have undertaken a more thorough analysis of the behavior of small consortia of macrophages in ligand distinction. Prompted by Reviewer 3 we have relaxed our assumptions about how consortia may function. Specifically, we redefined the probability of perfect distinction as the number of cells distinguishing each pair being at least twice the number of cells confusing the pair. Under this new calculation, the model consistently predicts poorer performance of consortia of cells in the I κ B α ^{S/S} mutant compared to WT (revised **Figure 6D**, Manuscript **Lines 482-491**). The difference is not big, but as described above, even in the prior experimental analysis the differences were not big, but sufficient to produce a subtle inflammatory phenotype.*

- Lack of experimental evidence for a major conclusion: Their model generates a prediction that individual cells can compensate for errors made by their brethren because the types of errors made are random. While I appreciate that a direct testing of this hypothesis is outside the scope of this study, it would be helpful for the authors to suggest – at least – an experiment that could be employed to test their hypothesis. If testing it is impossible, the conclusion will remain speculation.

*Reply: We thank the reviewer for the suggestion. The revised main conclusion emphasizes the diversity of single-cell stimulus-response specificity (scSRS) inferred from a mechanistic model parameterized based on experimental data. To test our major prediction regarding scSRS diversity, we have discussed potential experimental strategies to measure scSRS and test the predictions (**lines 573-581**). One such approach could be to isolate recently divided sibling cells and stimulate them with distinct ligands. This approach would face the technical challenge of isolating individual cells without perturbing their cell state (Chen et al., 2017). Another approach may be to learn how cell responses to one stimulus are indeed affected by prior stimulus (or develop better models to account for those changes). Then by tracking individuals during repeated stimulation regimes, the appropriate computational interpretation could approximate scSRS profiles for the tracked cells.*

Reviewer #1 (Remarks on code availability):

The code is very extensive, but appears well-organized and annotated enough to make the model usable to others in the field. It is transparently presented.

Reviewer #2 (Remarks to the Author):

The focus of this manuscript is in the broad area of NFkB signaling, a transcription factor known to be critical in immune cell response to stimuli, often pathogenic or cytokine driven. The manuscript advances our understanding of the single-cell stimulus-specific response through theoretical framework, beyond what is readily accessible through experimental data. The authors expand upon a previously published model from their group to generate a virtual cell and assess the NFkB dynamics in response to various stimuli. Of their main observations, they note that information loss is mainly due to extrinsic noise in the receptor module. Further, they conclude from their simulations that while there may be blind-spots, the heterogeneity in cell populations of as small as 2-6 macrophage consortia is sufficient to achieve perfect stimulus distinction. The manuscript is well written, with demonstration of good scientific rigor. However, there seems to be a lack in novelty or major conclusions in the manuscript which distinguishes the current draft from their previously published work. There are some concerns which could be addressed to enhance its clarity, impact and scholarly rigor.

We thank the reviewer for carefully assessing the manuscript and their appreciation that the math model-based approach allows for analyses not “readily accessible through experimental data” “with demonstration of good experimental rigor”. However, we would like to respectfully disagree about the lack of novelty or lack of major conclusions beyond what we previously published. What we previously published (Adelaja et al., 2021) was the experimental data and mathematical model that provided the simulated trajectory of one “representative cell” but not the actual heterogeneity of single cell data in the population, or a quantitative evaluation of the simulations. What Adelaja et al., (2021) provided was standard in the literature. With the new work we provide for the first time simulations of all single cells in the population that thereby capture the heterogeneity in the population. That allows us - for the first time - to address questions that are a function of the heterogeneity, i.e. stimulus responses specificity, and information loss. Then we introduce the novel concept of scSRS. Our analysis reveals that single cells differ remarkably in which stimulus pairs they are unable to distinguish, which leads to the conclusion that small consortia of just 2-6 macrophage may collaborate to achieve perfect stimulus distinction”

Major concerns:

1. This paper is mainly reliant on presenting a theoretical framework as a tool for generating synthetic data where experimental data is not easily attainable. However, given the model has already been published by the authors in 2021 (Adelaja et al., 2021), there is no new development in that front which distinguishes this manuscript from their previous work. Even the choice of parameters has been derived from previous studies of the authors and others.

*Reply: We agree with the reviewer that the present work is based on the NFkB signaling network model from Adelaja et al., (2021), which provided simulations matching one representative cell behavior for each stimulus condition. However, the key advancement in this work is the reconstruction of cellular heterogeneity observed in single-cell NFkB signaling responses using mechanistic model simulations under inferred parameter distribution. To clarify the novelty of our work, we have replaced **Figure 1A-B** with a workflow figure that delineates how the new work is quite distinct from prior work.*

Specifically, we first generated heterogeneous single-cell simulation:

- (1) We adaptively estimated parameter distributions by optimizing the fit to each single-cell trajectory using stochastic approximation expectation-maximization algorithm (SAEM). Adelaja et al., (2021) showed the simulation for one representative cell which suggested*

that the topology of the model is sufficient. Our work identified the parameter distribution, that provided a quantitative fit for the NFκB response distribution and accounts for the heterogeneous experimental data.

- (2) We quantitatively evaluated these fits and validated the heterogeneous dataset generated from model simulation. Given the complexity of the datasets we developed a series of QC metrics that establish a new standard for the field.

Compared to the previous one representative cell model (Adelaja et al., 2021), our heterogeneous single-cell models address, for the first time in NFκB literature, biological questions driven by cellular heterogeneity. We achieve this by developing novel workflows based on mechanistic model simulations of heterogeneous single-cell responses. These applications include:

- (1) In this work, the simulation of heterogeneous single-cell NFκB responses serve as a platform for **in-silico experiments**. We demonstrated examples of applying our model for extended dose predictions and IκBa mutant predictions. Adelaja et al., (2021) models one representative cell NFκB responses, and therefore could not reproduce heterogeneity within the population of macrophages.
- (2) We developed a novel approach to **identify the information loss** within the signaling network, by denoising different modules. Adelaja et al., (2021) could not undertake this analysis. In previous studies (Cheong et al., 2011; Cruz et al., 2021) the information loss has been assessed by experiments using different approaches. Our findings are consistent with these results validating our model.
- (3) Using the model validated in multiple ways, we were able to investigate the stimulus-response specificity of single cells (scSRS). This is an entirely novel analysis enabled by the model. It then allowed us **to investigate the heterogeneity of single-cells in their scSRS**. We simulated single-cell responses to different stimuli, which are experimentally infeasible due to the adaptive nature of macrophages (i.e. they cannot be stimulated multiple times without altering their responses). Our simulations reveal high diversity in scSRS, promoting stimulus-response specificity in small consortia of cells and allows the estimation of group size of achieving perfect distinction of stimuli.

These insights enabled by the mechanistic simulations of heterogeneous single-cell responses are the hallmarks of this paper. To clarify the advancement and the novelty of this work, we have revised introduction to explain previous models failing in capturing single-cell heterogeneity (lines 119-121), added workflow Figure 1A (lines 138-152 in introduction section), Figure 4B (lines 340-342 in results section) to clarify our developed approach to assess information flow (lines 552-558 in discussion section).

2. The authors suggest the current model to be more complex (133 parameters, 52 species) than some of the other works in the area (52 dimensions vs ~10 dimensions). Yet only a subset of biochemical constants is chosen for parameterization. (a) The authors should provide more rationale behind the choice of these parameters. (b) Is the model biased by the selection of these parameters, and how accurately does it reflect experimental data if different set or additional parameters were chosen? It appears the rest of the biochemical constants come from prior literature. (c) Are these interchangeable for the two cell descriptions of BMDM and hMPDMs, given their arguments in an earlier study about the heterogeneity of BMDM population as opposed to the hMPDMs?

Reply: Thanks for reviewer's carefully reading and constructive suggestions.

(a) We have provided more rationale of parameter choice in the Method section (Parameter selection and sensitivity analysis, Lines 625-638). We have appended the corresponding texts here:

“The single-cell parameters of the NFkB network ODEs were inferred from experimental datasets to recapitulate the heterogeneity (explained in detail in next section). To avoid overfitting, only a limited number of parameters were selected based on following four principles: (1) parameters with direct or strong biological evidence indicating heterogeneous measurements were prioritized, such as receptor abundances, trafficking capacities; (2) across different receptor module, consistent parameters are chosen, such as receptor synthesis rate; (3) redundancy in parameter functions was avoided—for instance, while both receptor synthesis and degradation rates can regulate receptor abundance, only one of them was selected, the receptor synthesis rate, as they alter NFkB dynamics in a similar way; (4) core module parameters were selected based on their capacity of generating oscillation heterogeneity.”

(b) As the reviewer correctly noted, other biochemical constants are based on prior literature, and we have clarified this in the revision (Methods, Lines 694). To evaluate whether parameter selection affected model performance, we have added estimation results for 42 different parameter sets in Figure S2M (Lines 274-275), which cost approximately 21,000 node hours. Our findings indicate that as more parameters are estimated, the fit quality generally increases (removing single parameters from estimated sets gives worse fit quality). The search space for optimal parameter set is extremely big, given that the model has 133 parameters, for example, the possible sets of more than 20 parameters in total will be around 10^{40} . Additionally, estimating each parameter set for each ligand condition requires around 500 node hours, which imposes practical constraints on exploring a larger range of parameter sets. We hope this explanation clarifies the rationale and limitations of our approach.

(c) We apologize for referencing hMPDMs – only BMDM datasets were used for the work described in this manuscript. BMDMs and hMPDMs are highly similar (almost indistinguishable) in contrast to the often-used Raw264.7 cell line (Singh et al., 2024, Figure S1A-C).

3. It is also unclear why the authors chose two different cells in this study and how some of their conclusions about the macrophage consortia would change if they utilized only one cell line. Furthermore, it is quite confusing where they utilized the hMPDMs experimentally, as stated in the methods section. From the main text, it appears that simultaneous ligands were introduced only for the simulation case and not experimentally (Fig5). The authors should revisit this and make it clear.

Reply: We apologize that hMPDMs applied to a different set of experiments that were included in a previous version of the paper - we forgot to remove the description after revising paper. Only BMDMs were used for the experimental studies that enabled the parameter estimation in this study. We have removed hMPDMs description.

4. (a) Expanding on the previous points, is the model universally applicable or do all the parameters need to be re-computed for another cell line/ model system? If the latter, then the process becomes tedious and limits the utility of the framework. (b) Furthermore, if sequencing experiments can provide similar inferences about NFkB dynamics as shown by the authors in their previous work (Sheu et al., 2023), isn't that a more direct and informative approach?

Reply: These are interesting questions and we will address them here:

(a) The model topology of the NFκB signaling network is conserved between cell types, and interaction and catalytic rate constants are generally genetically encoded. Expression parameters are determined by epigenetic mechanisms and so are cell type specific. This is beyond the scope of the present manuscript but was explored by one of our recent studies (Mitchell et al., 2023).

Our prediction for the IκBα^{S/S} genotype demonstrates the general predictive power of the model. With only a single parameter modification (NFκB-mediated IκBα transcription rate), the model was able to qualitatively predict increased ligand confusion in alignment with experimental observations, showing its robustness and adaptability.

(b) The present work is focused on mechanisms that control stimulus-specific NFκB signaling dynamics. Given the remarkable capability of a molecular mechanistic model we are able to investigate mechanisms and infer the SRS of single cells. Sheu et al., (2023) provides an alternate attempt to measure stimulus response specificity via scRNAseq, which is more inline with what is commonly done and is also more broadly applicable. However, this approach is limited in identifying molecular mechanisms that control SRS, nor can it reveal the SRS of single cells.

5. From Fig1C, it can be noted that the number of cells vary between 176 to 744 which is more than 4-fold different. Can the authors comment on how this impacts statistically on the model parametrization? Do the authors speculate a similar sample size for all the conditions may alter the model outcomes?

Reply: For each receptor module, all doses were used to estimate the parameters. For LPS and CpG two additional doses were previously also included in the model parametrization. Thus, the sample size for the parameter estimation of five ligands are: 1809 cells for TNF, 1715 cells for Pam, 1495 cells for LPS, 1558 cells for CpG, and 1763 cells for pIC. We have revised this information in Figure 1C and added the description of cell numbers used for parametrization for each condition in the Methods section (Lines 689-694).

To test the impact of a smaller sample size on parameter distribution, in the revision, we have redone the parameterization after down-sampling the Pam data to half size (see Figure S2H-L, Lines 268-274). We chose Pam because Pam are bacterial PAMPs and similar as CpG and LPS condition (these two conditions having smaller sample size). The results demonstrated that such down sampling did not change the fitting performance significantly. This supports the different sample sizes will not substantially influence the fitting performance and suggests that smaller sample size may not alter model outcomes.

6. Some of the choices seem subjective as they are reliant on visual inspection of trajectories. For eg. RMSDs threshold of 0.03 for acceptable fits

Reply: We acknowledged that choosing a threshold for RMSD is subjective, as it's hard to decide an objective threshold RMSD. We have revised the manuscript and replaced subjective description "RMSDs threshold of 0.03 for acceptable fits" with more objective description of the fitting results (Lines 218-219):

"Between 65% and 90% of the RMSDs for individual fitted trajectories under each condition are below the threshold of 0.03 (Figure S2B)"

In addition, we developed additional criteria to assess model fitting, including Wasserstein distance of signaling codons between simulation and experiments for same condition (similarity), and across different conditions (specificity), comparing classification of ligands for both experiments and simulation (cross experiments and simulation).

7. The authors compared the model simulation against the live cell macrophages using machine learning classification. What is the rationale behind only outputting the ligand information at high doses alone, and not test the other conditions?

Reply: We chose high-dose ligands because such conditions give highest stimulus response specificity (Adelaja et al., 2021). We have revised the manuscript to explain the rationale for this (lines 256-257). Just in case reviewer would be interested, we here have tested all three doses for 5 ligands, and it shows comparable stimulus confusion and distinction between experiments and model (see figure below).

Results from Adelaja et al. (2021)

C

	Stimulus Prediction (5x CV)														
TNF_0.3	49.3%	0.7%	0.6%	8.1%	0.2%	0.9%	11.8%	8.6%	0.6%	12.7%	2.6%		3.7%	1.3%	1.1%
TNF_3.3	1.1%	44.2%	24.9%	0.9%	3.2%	5.4%	1.6%	1.8%	0.7%	2.5%	3.4%	2.5%	5.0%	1.1%	1.6%
TNF_33		31.9%	38.8%	0.8%	1.3%	8.4%	1.6%	0.8%	0.9%	0.9%	5.0%	6.9%	1.9%	0.8%	0.6%
P3C4_1	17.7%	1.2%	0.9%	27.6%	2.1%	1.2%	11.8%	4.5%	1.9%	24.5%	0.9%		3.3%	2.1%	0.2%
P3C4_10	1.6%	3.5%	2.2%	2.7%	32.2%	0.8%	3.8%	8.4%	4.9%	4.9%	1.4%	8.2%	15.0%	1.9%	
P3C4_100	0.4%	6.1%	6.1%	0.2%	0.7%	42.0%	0.4%	0.2%	0.9%	10.0%	29.0%	0.2%		3.3%	
poly(i:C)_3.3	15.2%	2.5%	1.5%	13.0%	1.5%	0.3%	22.9%	7.1%	1.5%	16.4%	4.0%		7.7%	4.3%	1.9%
poly(i:C)_10	6.0%	2.5%	1.9%	3.8%	9.1%		14.2%	33.4%	0.3%	10.1%	3.5%		10.7%	3.2%	1.3%
poly(i:C)_100	1.5%	1.0%	0.5%	1.0%	9.0%	0.3%	2.1%	3.6%	67.7%	2.1%	5.7%	0.5%	4.1%	10.6%	1.3%
LPS_0.3	9.6%	1.9%	1.3%	18.2%	2.4%	0.9%	12.2%	10.5%	1.7%	31.6%	0.6%	0.2%	6.2%	2.1%	0.6%
LPS_10	2.1%	3.4%	2.1%	0.8%	3.3%	8.1%	2.4%	2.6%	2.6%	1.3%	49.3%	6.2%	3.1%	4.1%	8.6%
LPS_333	0.4%	1.0%	1.9%		0.2%	22.1%	0.2%	0.4%	0.8%	0.8%	4.0%	67.5%		0.6%	0.4%
CpG_33	5.1%	5.3%	4.8%	5.3%	9.8%		8.0%	8.0%	4.5%	9.1%	8.3%		22.1%	8.0%	1.9%
CpG_100	1.0%	1.3%	0.3%	0.8%	14.8%	0.3%	3.0%	5.3%	9.5%	1.0%	7.5%	0.3%	9.3%	38.7%	7.0%
CpG_1000	4.4%	1.0%	0.3%	0.7%	0.7%	2.0%	2.0%	2.7%	0.7%	1.0%	22.5%	1.7%	2.7%	7.2%	50.2%

Figure legend

Confusion matrices illustrating the classification precision of ligand identity for experimental data (Exp), and for simulated data (Model). The machine learning model (Random Forest) uses NFkB trajectory Signaling Codons as input to encode and predict the ligand information. Top panels are from this work, bottom are from (Adelaja et al., 2021). This shows that the model reproduces the ligand dose information contained in experimental NFkB trajectories.

8. (a) TNF condition seems to be most distinguishable in the ML classification in Fig 2D. It is understandable that TNF produces oscillatory profile which is distinguishable from other ligands. (b) However, it is also to be noted that the core module parameters were first fitted to TNF stimulated dataset. Could this have led to a bias in this regard?

Reply: We thank the reviewer for the careful reading and valuable question. (a) As shown in Fig 2D, TNF condition is most distinguished in the experimental data. The model fits well to experiments (Figure 2A and 2C), and the classification shows similar results with experiments, further validate the model's recapitulating the experiments.

(b) We agree with the reviewer that fitting the common core model first to TNF and Pam datasets has the potential to favor their fits. As pointed out the TNF condition is the only one that shows strong oscillations, while Pam non-oscillatory dynamics are representative to all four ligands. To ensure that the model captures TNF-induced oscillations TNF experimental data cannot be overshadowed by data from other ligands, we prioritized fitting to the TNF condition. However, it is important to note that the resulting fits are not worse for other ligands. We have revised the Methods description to add this explanation for core module parameters being first fitted to TNF and Pam datasets (Lines 687-689).

9. The results of Fig3A, while tedious, can be verified experimentally. The authors should demonstrate at least a few examples outside of the training set, where their theoretical predictions match with experimental observations.

Reply: We thank the reviewer for this suggestion. In the revision, we tested our model on an additional dose of 33ng/mL TNF and 333ng/mL LPS using published dataset (Adelaja et al., 2021), which are beyond the training set using very high doses. The model predictions aligned well with the experimental dataset (Figure S3E). The model effectively recapitulates Speed, Peak, Total, and Osc signaling codon distributions, while Duration and EvL are not well captured (Figure S3F). The average Wasserstein distances between model-predicted and experimental signaling codon distributions for these two conditions are minimal compared to their ligand-specific differences (Figure S3G). The simulations show some cells with longer durations that were not observed experimentally. This may suggest an additional post-induction attenuation mechanism at high doses that is not included in the model. We have included these data and their discussion in Figure S3E-G and manuscript Lines 291-297.

10. The explanation of the discrepancy of TNF MI between theoretical vs experimental observations remains unclear. Furthermore, the authors describe in discussions that the information transmission via duration and EvL signaling codons could be due to intrinsic noise within the I κ B α loop. This does not reflect in the current information loss schematic of the model. The authors need to address this section more clearly.

Reply: We thank the reviewer for the comments. We have attempted to describe our thinking more clearly. We have now updated the figure to clarify the discrepancy of TNF MI between simulation and experiments (FigureS3I-J, Lines 321-329). In simulations, which have no technical and intrinsic noise, the EvL codon for the low dose can be distinguished from the unstimulated condition, and the medium dose can be distinguished from the low dose as cells have more sustained oscillations (Figure S3I). However, in experiments, the unstimulated condition and low dose condition are not distinguishable, as the peak detected in low dose is similar as noise observed in unstimulated conditions (Figure S3J). An obvious key difference is that in the unstimulated condition, there is no noise in the simulated trajectories but there is in the experimental data.

We also note that the experimental TNF-induced oscillatory trajectories are more pronounced and contain more intra-trajectory variability in period than the simulation data. The intra-trajectory variability may diminish information transmission via Duration and EvL signaling codons. Such intra-trajectory variability is due to intrinsic noise within the I κ B α feedback loop that was described by (Kellogg and Tay, 2015).

These authors suggested that the intrinsic noise in the negative feedback loop (that causes this intra-trajectory variability) may expand the resonance frequency and hence the robustness of the TNF-induced oscillations. We now know that oscillations in response to TNF are important as they avoid nucleosome eviction and epigenetic reprogramming (Cheng et al., 2021; Kim et

al., 2022). Therefore we suggest that intrinsic noise within the negative feedback loop has a functional role to ensure the robustness of TNF-induced oscillations and cytokine vs PAMP distinction, even at the expense of the ability to distinguish different doses of this cytokine (Lines 530-540).

11. Can the authors suggest how would they hypothetically validate testing their conclusion about the loss of information in the receptor module experimentally?

Reply: We thank the reviewer for this question which indicates that we had not done a good job in describing all available data. Actually, our model prediction that the receptor module can be a major information loss is supported by previous literature. Cheong et al. (2011) used a model involving one receptor module (TNF) and two core modules (ATF2 and IKK -NF κ B) to theoretically demonstrate and experimentally validate that the information bottleneck occurring upstream in the TNF signaling pathway. This suggests receptor module as the bottleneck for information flow. Furthermore, Cruz et al. (2021) measured NEMO and RelA signaling within cell in response to IL-1 and TNF, finding a high correlation between NEMO and RelA signaling, which suggests little information loss in IKK-NF κ B core module. These experiments and their findings align with our prediction regarding the locating the information loss to receptor module. This was explained in the discussion (Lines 546-552), and we have also added this explanation in the Results section (Lines 361-363)

12. As brought up in an earlier point, the section describing Fig5 seems unclear. (a) For eg., which cells were used for the experiments vs simulations, (b) were the initial conditions same for the parameterized virtual cell, (c) how is the heterogeneity introduced etc.? Please address this section more to avoid confusion for the readers.

Reply: We thank the reviewer for this suggestion. We have added the workflow for generating heterogeneous single-cell response to diverse stimuli (Figure S5A) and revised the Methods section to provide more detailed information about the approach (Lines 786-816). Below are responses by points:

(a) The single-cell parameters estimated from experimental data were used for simulations and comparisons with experimental results (Figure 2). The inferred single-cell parameters were then used for bootstrapping sampling for generating heterogeneous simulations (Figure 3).

(b) The initial conditions for each cell were distinct, depending on their steady state under the corresponding single-cell mechanistic parameters.

(c) For method details of introducing heterogeneity, we've appended the revised manuscript here (Lines 786-816):

"Based on inferred single-cell parameters, we developed an approach to generate heterogeneous virtual single-cell NF κ B network parameters via bootstrapping sampling and core-module parameter matching, as described in further detail in the subsequent paragraphs. The generated heterogeneous single-cell NF κ B network parameters were applied to simulate the heterogeneous single-cell NF κ B signaling trajectories in response to different ligands using the ODE.

Since the single-cell parameters inferred from the experimental dataset include only one receptor module and the core module, the parameters for the other four receptor modules were missing and must be inferred to reconstruct the complete NF- κ B signaling network for each single cell (Figure S5A). To generate 1000 heterogeneous virtual single-cell NF κ B networks, our approach began by sampling 200 sets of parameters from those inferred from experimental dataset for the first ligand and core module. This allowed us to identify the first receptor module

parameters along with the core module for each virtual single cell. To determine the parameters in the other four receptor modules for the same individual cell, we assigned each receptor module parameters from the single cells that sharing the same core module parameters (or most similar). This process generated 200 heterogeneous single-cell NFκB signaling networks. We then repeated this workflow for each receptor module, using it as the initial sampling module in turn, until the process was completed and 1,000 virtual single-cell NFκB signaling networks were generated.

For example, we first sampled 200 single-cell parameters inferred from TNF-stimulated cells. For each sample of these 200 cells, we obtained the TNF module parameters along with core module parameters, but still needed the parameters for the LPS, CpG, Pam, and pIC receptors. To assign the LPS receptor parameters to each of these sampled 200 cells, we calculated the differences between the sampled core module parameters and the LPS-stimulated single-cell core module parameters, selecting the LPS-stimulated cell with the smallest difference and assigning its LPS receptor module parameter value to the sampled cell. We repeated this process for the CpG, Pam, and pIC receptors. This approach allowed us to generate 200 virtual single cells with heterogeneity across all five receptor modules, using TNF as the first sampled ligand. Then we repeated this for LPS as first sampled ligand, and so on.”

13. The generation of new data sets for this section relating to Fig5 again relied on high doses of the ligands. Why are other doses not considered in the analysis?

Reply: Similar as comment 7, we chose high-dose ligands because they maximize the stimulus-specificity of the resulting trajectories (Adelaja et al., 2021). We have revised the manuscript to explain the rationale for this (lines 256-257).

Minor concerns;

1. Check the references again as there are some inconsistencies in the citation formatting.

Reply: We have carefully revised the citation formatting.

2. The authors setup the introduction describing how there are outstanding questions that can't be addressed experimentally. However, it seems that only correlation of population level SRS with specificities of individual cells is unanswerable with experiments alone while the true channel capacity is already computed from experiments. In this regard, finer dose gradations can also be experimentally interrogated.

Reply: We agree with the reviewer that the single cell SRS inference is the primary finding presented here that cannot be addressed experimentally. We have revised the introduction to clarify this point (Lines 98-99):

“While some of these questions may be challenging to address experimentally, others, such as scSRS, may be nearly impossible.”

In addition, identifying information bottlenecks and identifying mechanisms that facilitate or limit information transmission, or that enable the generation of specific signaling codons cannot be achieved systematically with experiments.

3. Maybe mention the total number of biochemical constants used for parameterization in the main text (at least the range) for clarity rather than just reference to supplementary figures for this detail.

Reply: We thank the reviewer for the suggestion. We have revised the manuscript to include the total number of biochemical constants used for parameterization in the main text (lines 191-195).

Reviewer #3 (Remarks to the Author):

The manuscript by Guo et al. addresses the important question how immune cells can reliably process information about stimuli despite the heterogeneity in cellular signaling. This is done by combining live-cell imaging of the transcription factor NFkB in mouse macrophages, mathematical modeling using the established model of Adelaja et al 2021 and a comprehensive data analysis pipeline. The approach is used to demonstrate how well the cellular NFkB response can be distinguished between stimuli, how the stimulus response specificity (SRS) is limited by noise and where information loss due to noise occurs. The parametrized single cell models are finally used to evaluate the SRS of 1000 simulated cells that were not investigated experimentally. This overall highly interesting approach can serve as a very good example how mechanistic modelling combined with data analysis can predict cellular responses to stimuli, doses and perturbations not measured. However, there is a number of questions with respect to the limits of the approach and how far the results expand the insights of the preceding Adelaja et al. paper.

We thank the reviewer for the appreciative comments. We have addressed the questions posed in the point-by-point response below.

Major points:

- 1. (a) The mathematical modeling uses the model of Adelaja et al. 2021 and reparametrizes it based on extensive single cell data. The earlier paper already included single cell data measurement based on an identical workflow to identify stimulus-response specificity (SRS), (b) therefore it is not really clear what set the new approach apart from Adelaja et al. and what additional insights are gained. (c) The workflow scheme 1B is identical to Fig 1B in the earlier paper.

Reply: We agree with the reviewer that Figure 1 did not clarify the novelty of the work. This has been revised by adding workflow of this work (revised Figure 1A). To address the reviewer's question more specifically, our response is as follows:

- (a) *We agree that the present work is based on the NFkB signaling network model from Adelaja et al., (2021), which provided model simulations matching population-average (the representative cell) behavior for each stimulus condition. However, the key advance of the present work is the reconstruction of cellular heterogeneity observed in single-cell NFkB signaling responses using mechanistic model simulations under inferred parameter distribution. Though Adelaja et al., (2021) already included single-cell measurements, but their model only matches a single representative of each stimulus condition, as was the standard in the field. Because it was a single trajectory the fit was not quantitatively assessed.*
- (b) *To clarify the novelty, we have added Figure 1A and 4B, revised introduction, results, and discussion. The detailed response to this point can be found in response to Reviewer 2 point 1.*

(c) We agree that the experimental workflow is similar as Adelaja et al., (2021), thus, we have moved the previous-version Figure 1B to revised Figure S1A. The differences between our experiment setting up and in Adelaja et al., (2021) is that in this study we blocked the paracrine TNF to remove its effect on signaling in response to PAMPs, which is clarified in the revised Figure S1A and explained in the manuscript (lines 172-174).

- 2. Some formulations could be misinterpreted as a development of a new model (line 80 'development of mathematical model simulations', abstract 'we developed a mathematical model'), this should be clarified to 'model simulations' or 'simulations of the mathematical model'.

Reply: Thanks for the comments and suggestion. We have revised the abstract and introduction:

"we developed mathematical model simulations that capture the cellular heterogeneity of stimulus-responsive NF κ B dynamics and the SRS performance of the population." (Abstract lines 33-35)

"Here, Here, we advanced the NF κ B mathematical model simulations from one representative cell to all single cells within the population" (Introduction lines 138-139)

- 3. Given the highlighted importance of the negative feedback regulation it is not clear why the regulation by A20 that forms a second negative feedback loop within the IKK+NF κ B core system is not included into model. How would that feedback effect the NF κ B dynamics and signaling codons? This needs at least to be discussed based on the established role of that feedback on the dynamics of core models with A20 feedback (line 49-52).

Reply: That's a great suggestion because there is confusion in the literature. In the revision, we have discussed the effect of A20 feedback on NF κ B dynamics (Lines 107-111). Specifically, A20 transcription is regulated by NF κ B just like I κ B α , but its role is not in controlling the dynamics of NF κ B signaling. Instead, it was described to provide rheostat control and desensitization to subsequent signals (Son et al., 2021; Werner et al., 2008). The reasons cited are that A20 reaches up further into the pathway than I κ B α . It functions enzymatically rather than stoichiometrically, and it has a long half life. There is confusion in the field because early studies suggested that A20 contributes to turning off IKK directly, but in fact its effects are at the upstream signaling complex. This was confirmed by interpreting combined NEMO and NF κ B imaging data (Cruz et al., 2021). At the RIP1 containing signaling complex it is essential for controlling TNF-induced necroptosis (Oliver Metzger et al., 2020). As we do not undertake repeat stimulations in this work, there was no need to include the A20 feedback.

- 4. The introduction is not balanced according to the contribution of other groups in the field, these need to be acknowledged more carefully. Highly dynamic NF κ B behavior, in particular oscillations have first been shown by M Whites lab (Nelson et al Science 2004, Ashall et al. Science, 2009), also by Sun et al. Plos One 2009 and others (lines 18-22). Some of these references are cited later in the context of 'cell-to-cell heterogeneity casting doubt' but those references are missing in the reference list (while several of the group references occur even twice). Also the discussion of informative signaling features was started early by S Gaudets group (RE Lee et al. 2014 MolCell) and should be cited (line 26-28).

Reply: We appreciate the reviewer's point that we should take care to provide a balanced account in the literature. In the Introduction we first described studies of unmodified cells: "Earliest biochemical studies..." Such studies revealed oscillatory activity in some TNF

stimulation conditions but also different dynamic trajectories in other conditions. Subsequent live cell imaging studies required genetic modification of cells with either RelA-FP fusion proteins alone or RelA-FP and an NFκB-inducible IκBα expression plasmid. These indeed showed oscillatory trajectories and heterogeneity, which is why these studies were cited in this context. Sun et al 2009 was the first study that used a knock-in FP reporter for RelA, minimizing the genetic modification on the signaling system. The reviewer's comment suggests that this technical distinctions between biochemical and imaging studies are not broadly appreciated and may instead give an impression of bias. Therefore we have simplified the Introduction and conflated the two approaches:

“Biochemical and imaging studies found that the activity of the transcription factor NFκB is highly dynamic (Hoffmann et al., 2002; Nelson et al., 2004; Ashall et al., 2009; Sung et al., 2009; Tay et al., 2010)” (Lines 70-72)

“Only with large amounts of single-cell trajectory data available, specific dynamic features within the NFκB activation trajectories could be identified conveying information about the stimulus (Lee et al., 2014; Adelaja et al., 2021; Aqdas and Sung, 2023).” (lines 79-82)

• 5. (a) The generated experimental data set is described as ‘using five ligands at 3 doses with up to 800 single cell NFκB trajectories for each condition’ (line 111-114). The details in Figure 1C show that the cell numbers vary between 176 and 744 and should be described like that. (b) Moreover, the possible impact of different cell numbers in the different conditions, especially the lower numbers for LPS and CpG stimulation, should be discussed. (c) Additionally, the goodness-of-fit is described as ‘with a median of ~0.02 for each of the 15 stimulus conditions’ (line 150) and ‘a threshold of 0.03 provided for acceptable fits (Figure S2B), which were satisfied by ~80% of the trajectories’ (line 152-153). However, Figure S2A and S2B show, that the median ranges from 0.01 to 0.03 and the fraction of trajectories with acceptable fits range from 60% to ~95% for the different conditions. This should be described more precisely since the data shows a different goodness-of-fit for the individual conditions. Particularly, it seems that the TNF low condition can be much better reproduced than e.g. the LPS medium condition.

Reply: We thank the reviewer for these suggestions.

(a) *We have revised the manuscript: “with 176 to 744 single cell NFκB trajectories for each condition (Figure 1C)” (Line 177).*

(b) *For cell numbers of different condition, we addressed this in response to Reviewer 2, comment 5. Here we paste in the response: “For each receptor module, all doses were used to estimate the parameters. For LPS and CpG two additional doses were previously also included in the model parametrization. Thus, the sample size for the parameter estimation of five ligands are: 1809 cells for TNF, 1715 cells for Pam, 1495 cells for LPS, 1558 cells for CpG, and 1763 cells for pIC. We have revised this information in Figure 1C and added the description of cell numbers used for parametrization for each condition in the Methods section (Lines 689-694). To test the impact of a smaller sample size on parameter distribution, in the revision, we have redone the parameterization after down-sampling the Pam data to half size (see Figure S2H-L, Lines 268-274). We chose Pam because Pam are bacterial PAMPs and similar as CpG and LPS condition (these two conditions having smaller sample size). The results demonstrated that such down sampling did not change the fitting performance significantly. This supports the different sample sizes will not substantially influence the fitting performance and suggests that smaller sample size may not alter model outcomes.”*

(c) *We appreciate the reviewer's careful suggestion to precisely describe Figure S2A-B. We agree that low-dose TNF condition appears better than medium-dose LPS in this figure.*

In response, we have revised the manuscript (lines 214-219): “We quantified root mean square deviation (RMSD) between the experimental and simulated NFκB trajectories for every cell (See Methods Model performance evaluation), revealing RMSD distributions that ranged from 0 to 0.06, with a median between 0.01 and 0.03 across the 15 stimulus conditions (Figure S2A). Between 65% and 90% of the RMSDs for individual fitted trajectories under each condition are below the threshold of 0.03 (Figure S2B).”

- 6. For the data fitting a sequential approach is used. (a) What is the motivation for that? Does the sequence of selected data impact the results, e.g. have TNF and PAM data a higher weight in that approach because of their earlier selection? (b) What would be the difference to applying a multi-experiment fitting approach, e.g. Fröhlich et al. (<https://doi.org/10.1038/s41540-018-0079-7>)? (c) Moreover, the statement about the simulated trajectories that were used for the analyses (‘For TNF conditions, cells with simulated trajectories in the top 33% of the coefficient of variation (CV) were selected for further analysis, including evaluation of the model simulation and its applications.’, last sentence in the supplement, section 5.3) is unclear and need to be described in more detail and explained why only TNF conditions are included in the selection process.

Reply: We thank the reviewer for the valuable questions.

- (a) *Yes, we agree with the reviewer that fitting the common core model first to TNF and Pam datasets makes them having higher weights. The rationale for this strategy is that the TNF condition shows strong oscillations, and Pam non-oscillatory dynamics are representative to all four ligands. With this strategy we ensure that that the model captures TNF-induced oscillations and model fitting was not overwhelmed by data from other ligands. It is important to note that the resulting fits are not worse for ligand conditions that were not used for fitting the core module.*
- (b) *When we parameterized the core module with all datasets (as in Fröhlich et al. 2018), then core module parameters fit the non-oscillatory dataset well but did not fit the oscillatory content of the TNF condition (which comprises only ~20% of the data).*
(a-b) We have revised the Methods description to add this explanation for core module parameters being first fitted to TNF and Pam datasets (Lines 687-689)
- (c) *We have also revised the supplementary materials (Section 5.3, last paragraph) to clarify the reason for selecting a subset of the fits from the TNF condition “For TNF conditions, due to the poor recovery of noisy sustained oscillations, the model simulations with high coefficient of variation (top 33% of CV) were selected for further analysis and evaluation.”*

- 7. While the statement that 80% of single cells distinguish 9 to 10 stimulus pairs is highly intriguing, it strongly depends on the chosen threshold. In Figure 5C the distribution of L2 distances between the given signaling codons is shown and two examples of corresponding trajectories. However, it is unclear how strong the difference between individual signaling codons is and more importantly how this translates to the transcriptional response.

Reply: We thank the reviewer for their interest in this finding. We agree that the reported ratio is dependent on the threshold applied and we present the results for different threshold (Figure S6C). This shows that the main finding the single-cell stimulus response specificity profile is heterogeneous is robust across different thresholds.

For the reviewer’s interest, we have analyzed the differences between signaling codons (see figure below). There is a wealth of subtle observations in this data that may go beyond the general interest of most readers. We agree that it is unclear to what extent these differences

result in differences in gene expression, and this remains an open question until experimental technology is improved.

Figure legend:

Left panel: Violin plots depicting the distribution of *l2* distances in each signaling codon space (signal codons labeled on the left) between the 10 possible stimulus pairs (specified in the x-axis) for WT. 10th, 50th, and 90th percentiles are marked from bottom to top within each violin plot.

Right panel: Violin plots depicting the distribution of *l2* distance in each signaling codon space (signal codons labeled on the left) between stimulus pairs (specified in the x-axis), for WT and *IkBα^{S/S}*. 10th, 50th, and 90th percentile are marked from bottom to top within each violin plot.

- 8. Finally, the authors claim that ‘neighboring cells compensate for each other’s blind-spots’ (line 351-352). This statement seems too strong given that in a multicellular context, one expects that the cellular response in this environment is a combination of the responding cells,

especially in the context of cytokine excretion. Cells that cannot distinguish two signals, could therefore dilute the response of cell distinguishing the signals. This should be clarified.

Reply: We thank the reviewer for this valuable and insightful comment and suggestion. We totally agree that the rules governing how consortia of cells function are critical for this thought experiment. We considered that cells that see distinctions can override cells that do not. Alternatively, cells that confuse stimuli could dilute the stimulus response specificity of the consortium if they retain a vote. Finally, if cells that confuse stimuli have veto power, the consortium would always fare poorly in stimulus response specificity. We have now discussed this assumption in the revision (Results Lines 461-463, Discussion Lines 589-592), and included an analysis that assume consortia governed by a 2/3 majority rule, i.e. consortia achieve perfect ligand distinction are defined as those where every ligand pair is distinguished by at least 2/3 of the number of cells in the consortium. Interestingly, in this new analysis the lower diversity consortia show much poorer performance on perfection distinction than the WT consortia which have more scSRS diversity.

In the revision, to make our conclusion more precise, the main conclusion have been changed to emphasizing the diversity of scSRS and its role in immune response. Such diversity allows cells to compensate for each other's blind spots, thereby achieving perfect distinction as a small consortium. While the size of the consortium depends on specific thresholds, the fundamental conclusion remains unchanged: without diversity, a consortium of cells would not be able to achieve perfect distinction. We have recalculated Figure 6C and added Figure S6C, and revised the corresponding manuscript (lines 472-480).

• 9. How does the block of para- and autocrine TNF signaling impact the results on small consortia of macrophages were the authors conclude that diversity among the single cells of a consortia enables perfect SRS? A discussion how far the data and analysis represent a case of modified interactions and diversity is required.

Reply: This is an interesting question. Prior studies of addressed the role of para- and autocrine TNF signaling on cell responses to PAMPs (Caldwell et al., 2014; Adelaja et al., 2021). It was noted that paracrine TNF can lead to oscillatory NFκB trajectories in cells that do not respond directly to the PAMP (Adelaja et al., 2021). Thus paracrine TNF signaling may diminish stimulus-distinctions at lower PAMP doses when the activation threshold is not met in every cell. Here we considered high doses because the SRS is higher. We do not know how paracrine TNF would affect the functioning of consortia or affect their governance rules.

10. There is information missing:

- table S1 und S2
- Fig 5I is missing

Reply: We thank the reviewer for the detailed and careful reading our manuscript. We have added table S1 and S2. We apologize that Figure 5I was mis-cited and have revised the corresponding manuscript (Lines 437-440). "The L2 differences revealed increased confusion between TNF vs. PAMPs (7.6% of 1000 cells), among bacterial PAMPs (18.3% of 1000 cells), and between bacterial and viral PAMPs (1.7% of 1000 cells) (Figure 5F, 5H)."

Minor points

- References to figures are incorrect in line 152 and 153: reference to Figure S2B should be Figure S2A, right panel, and the reference to Figure S2C should be Figure S2B.

Reply: We highly appreciate the reviewer's careful checking. We have revised the manuscript as the reviewer suggested (Lines 216-219).

- The statement in line 328 'for all stimulus pairs the 50th and 90th percentile of L2 were diminished' is incorrect. For CpG vs Pam it is increased.

Reply: We have revised the manuscript to fix this (Line 435). "The 50th percentile of the L2 distance distributions were diminished for all 10 stimulus pairs; 90th percentiles were lower for 9 pairs; the 10th percentiles were lower for 6 of 10 pairs."

- The statement 'The I κ B α S/S genotype reduced the proportion of cells being able to distinguish all stimulus pairs by 5% compared to WT' (line 330) is unclear. Is the 5% reduction related to the average of all conditions? The following given percentages in line 331-332 are also not clear where they come from. Because comparing the percentages of individual conditions for the I κ B α S/S genotype and WT show that for the I κ B α S/S genotype confusion can also be reduced e.g. for pIC vs PAM (16% to 13.1%).

Reply: In the corresponding paragraph pointed out by the reviewer, all percentages are related to 1000 cells (i.e. 5% = 50 cells / 1000 cells). We have revised the manuscript to clarify this (Lines 436-440). "The I κ B α ^{S/S} genotype reduced the proportion of cells being able to distinguish all stimulus pairs by 5% (of 1000 cells) compared to WT (Figure 5E, 5G). The L2 differences revealed increased confusion between TNF vs. PAMPs (7.6% of 1000 cells), among bacterial PAMPs (18.3% of 1000 cells), and between bacterial and viral PAMPs (1.7% of 1000 cells) (Figure 5F, 5H)."

- Figure number missing in supplement in section 6.1

Reply: We have revised the Supplement material to fix this.

Reviewer #3 (Remarks on code availability):

The code of the modeling and analysis is available and the documentation in the supplement is comprehensive.

Reviewer #4 (Remarks to the Author):

Reference

- Adelaja, A., Taylor, B., Sheu, K.M., Liu, Y., Luecke, S., Hoffmann, A., 2021. Six distinct NF κ B signaling codons convey discrete information to distinguish stimuli and enable appropriate macrophage responses. *Immunity* 54, 916–930.e7. <https://doi.org/10.1016/j.immuni.2021.04.011>
- Aqdas, M., Sung, M.-H., 2023. NF- κ B dynamics in the language of immune cells. *Trends Immunol.* 44, 32–43. <https://doi.org/10.1016/j.it.2022.11.005>
- Ashall, L., Horton, C.A., Nelson, D.E., Paszek, P., Harper, C.V., Sillitoe, K., Ryan, S., Spiller, D.G., Unitt, J.F., Broomhead, D.S., Kell, D.B., Rand, D.A., Sée, V., White, M.R.H., 2009. Pulsatile Stimulation Determines Timing and Specificity of NF- κ B-Dependent Transcription. *Science* 324, 242–246. <https://doi.org/10.1126/science.1164860>
- Caldwell, A.B., Cheng, Z., Vargas, J.D., Birnbaum, H.A., Hoffmann, A., 2014. Network dynamics determine the autocrine and paracrine signaling functions of TNF. *Genes Dev.* 28, 2120–2133. <https://doi.org/10.1101/gad.244749.114>
- Chen, Y.-C., Baac, H.W., Lee, K.-T., Fouladdel, S., Teichert, K., Ok, J.G., Cheng, Y.-H., Ingram, P.N., Hart, A.J., Azizi, E., Guo, L.J., Wicha, M.S., Yoon, E., 2017. Selective Photomechanical Detachment and Retrieval of Divided Sister Cells from Enclosed Microfluidics for Downstream Analyses. *ACS Nano* 11, 4660–4668. <https://doi.org/10.1021/acsnano.7b00413>
- Cheng, Q.J., Ohta, S., Sheu, K.M., Spreafico, R., Adelaja, A., Taylor, B., Hoffmann, A., 2021. NF- κ B dynamics determine the stimulus specificity of epigenomic reprogramming in macrophages. *Science* 372, 1349–1353. <https://doi.org/10.1126/science.abc0269>
- Cheong, R., Rhee, A., Wang, C.J., Nemenman, I., Levchenko, A., 2011. Information transduction capacity of noisy biochemical signaling networks. *Science* 334, 354–358. <https://doi.org/10.1126/science.1204553>
- Cruz, J.A., Mokashi, C.S., Kowalczyk, G.J., Guo, Y., Zhang, Q., Gupta, S., Schipper, D.L., Smeal, S.W., Lee, R.E.C., 2021. A variable-gain stochastic pooling motif mediates information transfer from receptor assemblies into NF- κ B. *Sci. Adv.* 7, eabi9410. <https://doi.org/10.1126/sciadv.abi9410>
- Hoffmann, A., Levchenko, A., Scott, M.L., Baltimore, D., 2002. The I κ B-NF- κ B signaling module: temporal control and selective gene activation. *Science* 298, 1241–1245. <https://doi.org/10.1126/science.1071914>
- Kellogg, R.A., Tay, S., 2015. Noise facilitates transcriptional control under dynamic inputs. *Cell* 160, 381–392. <https://doi.org/10.1016/j.cell.2015.01.013>
- Kim, J., Sheu, K.M., Cheng, Q.J., Hoffmann, A., Enciso, G., 2022. Stochastic models of nucleosome dynamics reveal regulatory rules of stimulus-induced epigenome remodeling. *Cell Rep.* 40, 111076. <https://doi.org/10.1016/j.celrep.2022.111076>
- Lee, R.E.C., Walker, S.R., Savery, K., Frank, D.A., Gaudet, S., 2014. Fold-change of nuclear NF- κ B determines TNF-induced transcription in single cells. *Mol. Cell* 53, 867–879. <https://doi.org/10.1016/j.molcel.2014.01.026>
- Mitchell, S., Tsui, R., Tan, Z.C., Pack, A., Hoffmann, A., 2023. The NF- κ B multidimer system model: A knowledge base to explore diverse biological contexts. *Sci. Signal.* 16, eabo2838. <https://doi.org/10.1126/scisignal.abo2838>
- Nelson, D.E., Ihekweba, A.E.C., Elliott, M., Johnson, J.R., Gibney, C.A., Foreman, B.E., Nelson, G., See, V., Horton, C.A., Spiller, D.G., Edwards, S.W., McDowell, H.P., Unitt, J.F., Sullivan, E., Grimley, R., Benson, N., Broomhead, D., Kell, D.B., White, M.R.H., 2004. Oscillations in NF- κ B Signaling Control the Dynamics of Gene Expression. *Science* 306, 704–708. <https://doi.org/10.1126/science.1099962>

- Oliver Metzigg, M., Tang, Y., Mitchell, S., Taylor, B., Foreman, R., Wollman, R., Hoffmann, A., 2020. An incoherent feedforward loop interprets NF κ B/RelA dynamics to determine TNF-induced necroptosis decisions. *Mol. Syst. Biol.* 16, e9677. <https://doi.org/10.15252/msb.20209677>
- Sheu, K.M., Guru, A.A., Hoffmann, A., 2023. Quantifying stimulus-response specificity to probe the functional state of macrophages. *Cell Syst.* 14, 180-195.e5. <https://doi.org/10.1016/j.cels.2022.12.012>
- Singh, A., Sen, S., Iter, M., Adelaja, A., Luecke, S., Guo, X., Hoffmann, A., 2024. Stimulus-response signaling dynamics characterize macrophage polarization states. *Cell Syst.* 0. <https://doi.org/10.1016/j.cels.2024.05.002>
- Son, M., Wang, A.G., Tu, H.-L., Metzigg, M.O., Patel, P., Husain, K., Lin, J., Murugan, A., Hoffmann, A., Tay, S., 2021. NF- κ B responds to absolute differences in cytokine concentrations. *Sci. Signal.* 14, eaaz4382. <https://doi.org/10.1126/scisignal.aaz4382>
- Sung, M.-H., Salvatore, L., Lorenzi, R.D., Indrawan, A., Pasparakis, M., Hager, G.L., Bianchi, M.E., Agresti, A., 2009. Sustained Oscillations of NF- κ B Produce Distinct Genome Scanning and Gene Expression Profiles. *PLOS ONE* 4, e7163. <https://doi.org/10.1371/journal.pone.0007163>
- Tay, S., Hughey, J.J., Lee, T.K., Lipniacki, T., Quake, S.R., Covert, M.W., 2010. Single-cell NF-kappaB dynamics reveal digital activation and analogue information processing. *Nature* 466, 267–271. <https://doi.org/10.1038/nature09145>
- Uda, S., Saito, T.H., Kudo, T., Kokaji, T., Tsuchiya, T., Kubota, H., Komori, Y., Ozaki, Y., Kuroda, S., 2013. Robustness and compensation of information transmission of signaling pathways. *Science* 341, 558–561. <https://doi.org/10.1126/science.1234511>
- Werner, S.L., Kearns, J.D., Zadorozhnaya, V., Lynch, C., O’Dea, E., Boldin, M.P., Ma, A., Baltimore, D., Hoffmann, A., 2008. Encoding NF-kappaB temporal control in response to TNF: distinct roles for the negative regulators I κ B α and A20. *Genes Dev.* 22, 2093–2101. <https://doi.org/10.1101/gad.1680708>

REVIEWERS' COMMENTS

Reviewer #1 (Remarks to the Author):

The authors have reasonably addressed all of my previous claims. I still argue that the progress past their past paper is somewhat incremental, but I think they were responsive to everything raised by myself.

We thank the reviewer for their comments and for appreciating our responses. We agree that the development of the model may be viewed as incremental, but would like to suggest that the characterization of single-cell Stimulus-Response-Specificity (scSRS) is conceptually innovative, involved new quantitative analysis tools, and that our analysis reveals a previously unknown characteristic of heterogeneous macrophage functions.

Reviewer #2 (Remarks to the Author):

The authors have amended fig1 and corresponding text in the manuscript to highlight the differences. However, it is evident that the underlying model is the same framework used in Adelaja et al. (2021). In the previous work, the fit is done on a representative profile (the choice of the representative case is not clearly described) while this work considers the different profiles which represents the heterogeneity. In the rebuttal, the authors mention it was the norm to only model a representative profile but other studies model either using an average expression profile (by the authors themselves in DOI: 10.1126/science.1071914) or some that have also considered single-cell data (DOI: 10.1126/science.1164860). Moreover, several studies have demonstrated single cell models with heterogeneity through different parameter selection (doi.org/10.1038/nature09145, doi.org/10.1038/ncomms12057, doi.org/10.1073/pnas.0913798107, doi.org/10.1186/1752-0509-7-45), hence the claim of it being done for the first time in this manuscript, is misleading. Further, the authors acknowledge that first fitting the core module to TNF and PAM may favor their fits. In other words, does it accurately capture single cell heterogeneity in cellular response. It is mentioned that fits are not worse for other ligands by this approach but is not shown.

We thank the reviewer for the further comments, and we have clarified our statement: it has been the norm in the NF κ B field where the dynamical model contains dozens of parameters to present a model that is parameterized to a representative trajectory that is either the average or selected in some other way. That is true for Hoffmann et al 2002 and also Adelaja et al 2021. Examining the other papers that were mentioned, we find that:

Ashall et al 2008 Science: measured single-cell NF κ B response to TNF stimulation and introduced the cellular heterogeneity into the NF κ B model by incorporating gene expression stochasticity. 5 cells from sim and experiments are displayed for comparison (see below). The authors suggested that there is a qualitative fit. However, it is not clear to what extent the model simulations fit quantitatively and whether these fits may be extended to more cells or larger populations of single cells.

Tay et al 2010 Nature: presents single-cell experimental data. The model simulations capture some aspects of the experimental data (Fraction of active cells, distributions of total NF κ B activity, response time). However, similar as last paper, 5-6 cells from experiment and 10 from simulation are displayed (see below). It is not clear to what extent the model simulations fit quantitatively and whether these fits may be extended to more cells or larger populations of single cells. The fits are not evaluated quantitatively and correlations between features remain unknown.

Adamson et al 2016 Nature Comm. Presents single cell experimental data. Model simulations capture some aspects of the experimental data (refractory period), but the timecourse trajectories are not presented side-by-side and the fits are not evaluated quantitatively.

Paszek et al 2010 PNAS. Presents bulk experimental data and the concept that population robustness arises from cellular heterogeneity. This argument is made with model simulations in which parameters are distributed to model single cell heterogeneity. However, the resulting simulation trajectories (see below) are not compared to or fit to single-cell experimental data. Similar theoretical arguments, though less extensive, were made in Nelson et al 2004 Science and Barken et al 2005 Science.

Joo et al 2013 BMC Systems Biology. Uses previously published bulk experimental data and model simulations in which parameters are distributed. However, the resulting trajectories are not compared to or fit to single-cell experimental data. So the study remains theoretical.

There is one previous paper that compared simulated and experimentally determined trajectories. That is Cheng et al 2015 Science Signaling. This paper matches trajectories generated by experiments with those generated by a mathematical model, however the fits are not quantitatively evaluated and the parameters are not optimized for an optimal fit.

In sum, this discussion suggests to us that we should include a more careful description (Lines 94-112) of the prior literature in the text, which we have now included in Introduction.

Reviewer #3 (Remarks to the Author):

We thank the authors for their answers, modifications and additional analyses, which explain and clarify many aspects of the manuscript. Most of the major points are now resolved. In particular, the modified Fig 1a and the revised text parts clarify what is new compared to Adelaja et al. (2021) (point 1).

Point 7. The presentation of the investigation for a different threshold shows the robustness of the results. The reviewer agrees that the additional data provide a wealth of information and it would be highly interesting how this translates into gene expression.

Point 8. The clarification of the underlying assumptions makes the line of argumentation now more clear.

We thank the reviewer for this further evaluation and appreciation for our clarifications.

There are three open points

Point 3. The authors argue that the A20 transcriptional feedback is not important for the early NFkB response. While this has been demonstrated for short time frames (Cruz et al., 2021), it might not be correct at later time points, as Werner et al., 2008 showed that a A20 KO can change the single cell dynamics of decaying NFkB after around 60 minutes for TNFa constantly present. Since in the manuscript single cells are partly tracked until 8h, A20 might have an impact depending on how long TNF is present in the experiment and the model. It would be good to clarify this point.

We thank the reviewer for pointing this out. Yes, it is true that A20 could play a role in dampening the late times. We actually produced experimental data from macrophages produced from a reporter mouse in the A20 cre-loxed background, but the results were not clear (data remains unpublished, in part because we do not know for sure that the cre excision was successful and the macrophages were actually A20-deficient). (The Werner et al 2008 work was in MEFs.) Still, it is possible that A20 has an effect in macrophages and that our model involves compensatory parameter values to make up for the lack of A20 in the model. We have included this point in the Discussion (Lines 488-490).

Point 4. A reference list is missing in the revised paper version.

We apologize for this oversight.

Point 10. There are still no tables S1 and S2.

We apologize for this oversight.

Reviewer #4 (Remarks to the Author):
